# LuxDiT: Lighting Estimation with Video Diffusion Transformer

**Ruofan Liang**[1,2,3*]   **Kai He**[1,2,3]   **Zan Gojcic**[1]   **Igor Gilitschenski**[2,3]

**Sanja Fidler**[1,2,3]   **Nandita Vijaykumar**[2,3†]   **Zian Wang**[1,2,3†]

[1]NVIDIA   [2]University of Toronto   [3]Vector Institute

## Abstract

Estimating scene lighting from a single image or video remains a longstanding challenge in computer vision and graphics. Learning-based approaches are constrained by the scarcity of ground-truth HDR environment maps, which are expensive to capture and limited in diversity. While recent generative models offer strong priors for image synthesis, lighting estimation remains difficult due to its reliance on indirect visual cues, the need to infer global (non-local) context, and the recovery of high-dynamic-range outputs. We propose LuxDiT, a novel data-driven approach that fine-tunes a video diffusion transformer to generate HDR environment maps conditioned on visual input. Trained on a large synthetic dataset with diverse lighting conditions, our model learns to infer illumination from indirect visual cues and generalizes effectively to real-world scenes. To improve semantic alignment between the input and the predicted environment map, we introduce a low-rank adaptation finetuning strategy using a collected dataset of HDR panoramas. Our method produces accurate lighting predictions with realistic angular high-frequency details, outperforming existing state-of-the-art techniques in both quantitative and qualitative evaluations. Project page: https://research.nvidia.com/labs/toronto-ai/LuxDiT/

## 1   Introduction

In physically-based rendering, lighting plays a central role in shaping the appearance—how objects cast shadows, reflect, and appear integrated within a scene. From virtual object insertion and augmented reality to synthetic data generation, many downstream tasks rely on estimating scene illumination. Yet inferring lighting from casually captured images or video remains an open challenge.

A common representation of the scene illumination is the high-dynamic-range (HDR) environment map, which describes incoming light intensity from all directions. HDR maps can be acquired by using light probes or multi-exposure panoramas, requiring specialized setups that are impractical for everyday use [9]. To overcome this, several learning-based methods that estimate environment maps directly from casually captured LDR images or videos have been proposed [15, 16, 33, 74]. However, these methods typically depend on paired datasets of input images or videos and HDR environment maps, leading to a chicken-and-egg problem: a large collection of HDR environment maps is needed to train a model that aims to alleviate the need for acquiring such expensive data in the first place.

Recently, generative diffusion models have demonstrated strong capabilities in modeling complex image distributions. DiffusionLight [45] demonstrated that pretrained text-to-image models encode implicit knowledge of illumination, which can be cleverly extracted by inpainting a virtual chrome ball into an image, generating plausible appearances under varying exposure settings. However,

---

* Work done during internship at Nvidia

† Joint Advising

39th Conference on Neural Information Processing Systems (NeurIPS 2025).

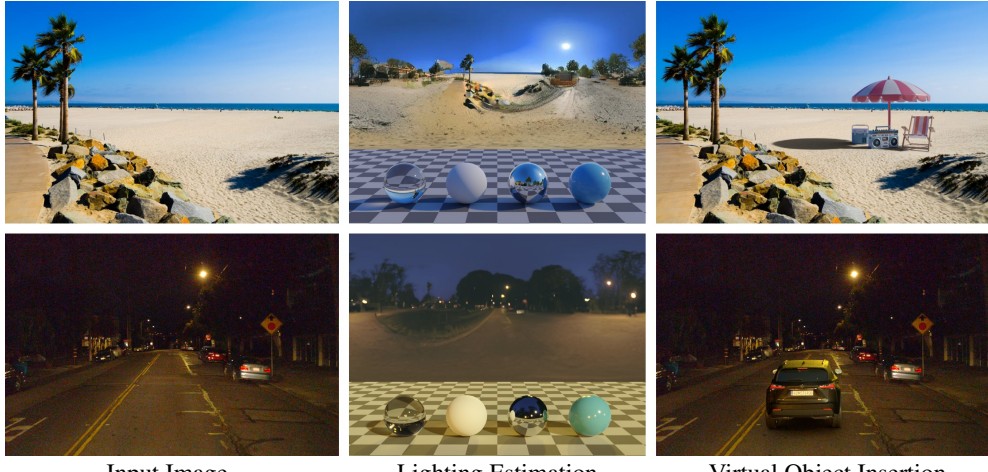

| Input Image | Lighting Estimation | Virtual Object Insertion |
|---|---|---|

Figure 1: LuxDiT is a generative lighting estimation model that predicts high-quality HDR environment maps from visual input. It produces accurate lighting while preserving scene semantics, enabling realistic virtual object insertion under diverse conditions.

without task-specific fine-tuning, the inpainting priors of pre-trained diffusion models are insufficient for producing reliable lighting estimates in a single inference and cannot directly generate HDR outputs. As a result, DiffusionLight relies on an expensive test-time ensemble strategy to improve robustness. Moreover, sampling multiple exposures through separate inference passes introduces inconsistencies and limits the dynamic range of the reconstructed illumination.

In this work, we formulate lighting estimation as a conditional generative task and propose LuxDiT, a neural lighting predictor trained on synthetic data and adapted to real-world scenes. Conditioned on visual input, our approach fine-tunes a diffusion transformer (DiT) to synthesize HDR panoramas from noise. Unlike pixel-aligned tasks, lighting estimation requires global reasoning over scene context. DiTs are particularly suited to this task: their attention-based architecture supports global context aggregation, and their generative priors facilitate reasoning from indirect cues such as shading and reflections.

Training such a model requires diverse lighting data. To overcome the lack of real-world HDR lighting supervision, we construct a large-scale synthetic dataset with randomized geometry, materials, and lighting conditions. Training on this dataset allows the model to learn physically grounded cues for light direction and intensity. While this imparts general lighting priors, models trained purely on synthetic data often hallucinate lighting based on dataset priors, producing environment maps that are plausible but semantically mismatched with the input scene. For example, an image of an urban street may yield an environment map depicting a rural landscape. To address this, we further apply low-rank adaptation (LoRA) [24] on a curated set of real HDR panoramas, improving alignment between predicted lighting and scene semantics.

Given a single image or video, LuxDiT produces HDR environment maps with accurate direction, intensity, and scene-consistent content. It reduces lighting estimation error by 45% on Laval Outdoor sunlight direction and improves temporal consistency for video input, enabling reliable use in downstream applications such as virtual object insertion. Our main contributions are:

- A DiT-based generative architecture that synthesizes HDR environment maps from visual input.
- A LoRA-based fine-tuning strategy using curated HDR panoramas to improve semantic alignment between the input scene and predicted illumination.
- A large-scale synthetic dataset with randomized geometry, materials, and lighting.

## 2 Related Work

**Lighting estimation** aims to infer environment illumination from input imagery, and is critical for photorealistic rendering and virtual object insertion. Early learning-based methods treat lighting estimation as a supervised regression problem, predicting spherical lobes [16, 33, 72, 69], parametric sources [66, 14], or low-resolution environment maps [15, 50, 74, 52] directly from a single image. These models are trained on paired data obtained from real-world captures [15, 50, 58] or synthetic

rendering [33, 74, 52]. However, their performance often degrades in complex, in-the-wild scenes due to limited diversity in the training data.

Recent methods incorporate generative priors to address the ambiguity of scene illumination. Style-Light [55] fine-tunes a StyleGAN to generate LDR and HDR panoramas from latent codes, using GAN inversion at test time. However, its performance hinges on inversion quality and often breaks semantic alignment on out-of-domain inputs. EverLight [8] regresses a parametric lighting estimate and refines it with a GAN to add high-frequency detail, but relies on pseudo-labeled HDR data and struggles with complex or bright lighting. DiffusionLight [45] uses a diffusion model to inpaint a virtual chrome ball under multiple exposures, merging them into an HDR map. While visually plausible, this multi-stage process yields distorted panoramas and limited dynamic range.

**Inverse rendering** recovers scene properties such as geometry, material reflectance, and illumination from image observations. Lighting estimation is often treated as a subcomponent of this broader task, with prior work jointly estimating lighting alongside depth, normals, and albedo. Learning-based approaches [49, 33, 59] typically leverage physics-based constraints and use re-rendering losses to supervise predictions. However, these methods often assume simplified reflectance models such as Lambertian shading, which limits their ability to handle complex lighting effects.

Optimization-based methods leverage differentiable rendering [4, 71, 70, 6, 60, 42, 18, 34] to jointly optimize lighting parameters and other scene attributes through photometric losses and regularization terms. Some approaches [31] follow a decomposition-then-optimization strategy: estimating geometry and albedo first, then solving for lighting via optimization. Other works also explore priors from proxy geometry [65] or pretrained general models [38, 36, 43]. The optimization-based pipelines often require dense multi-view captures or known proxy geometry, and involve expensive test-time optimization procedures. In contrast, our method directly predicts HDR illumination in a feed-forward manner without requiring scene geometry or iterative inference.

**Diffusion model priors.** Diffusion models (DMs) have emerged as a powerful class of generative models in high-fidelity image [46, 2, 47, 7] and video synthesis [22, 73, 3, 63, 1]. Beyond generation, pretrained DMs have been adapted to perception tasks through task-specific finetuning on carefully curated datasets [62, 39, 19], showing strong results on spatially aligned predictions such as depth [30, 25, 29], surface normals [13, 64, 35], albedo [11, 31, 68, 35], and material properties [31, 68, 35, 43]. Adapting DMs to non-local tasks like lighting introduces new modeling challenges, as outputs such as HDR panoramas are not spatially-aligned with the input.

## 3 Preliminaries: Diffusion Models

Diffusion models learn to approximate a data distribution $p_{\text{data}}(\mathbf{x})$ through iterative denoising. Following DDPM [21], a forward process progressively adds Gaussian noise to a data sample $\mathbf{x}_0 \sim p_{\text{data}}$, producing a noisy version at timestep $t \in [1, T]$ as: $\mathbf{x}_t = \sqrt{\bar{\alpha}_t}\mathbf{x}_0 + \sqrt{1 - \bar{\alpha}_t}\epsilon$, where $\epsilon \sim \mathcal{N}(\mathbf{0}, \mathbf{I})$ and $\bar{\alpha}_t$ defines the noise schedule. During training, a neural network $\boldsymbol{\mu}_\theta$ learns to reverse this process by minimizing:

$$\mathbb{E}_{\mathbf{x}_0 \sim p_{\text{data}}(\mathbf{x}), t \sim p_t, \epsilon \sim \mathcal{N}(\mathbf{0}, \mathbf{I})} \left[ \|\boldsymbol{\mu}_\theta(\mathbf{x}_t; \mathbf{c}, t) - \mathbf{y}\|_2^2 \right], \tag{1}$$

where $\mathbf{c}$ represents optional conditioning inputs. The denoising target $\mathbf{y}$ varies by formulation, and can be the noise $\epsilon$ [21], the v-prediction $\sqrt{\bar{\alpha}_t}\epsilon - \sqrt{1 - \bar{\alpha}_t}\mathbf{x}_0$ [48], or the clean signal $\mathbf{x}_0$ itself [28]. At inference time, samples are generated by denoising an initial Gaussian sample through a fixed number of reverse steps. In this paper, we build on CogVideoX [63], a latent video diffusion model trained on compressed video representations. A pretrained auto-encoder pair $\{\mathcal{E}, \mathcal{D}\}$ maps RGB videos to and from a latent space, such that $\mathcal{E}(\mathbf{x}) = \mathbf{z}$ and $\mathcal{D}(\mathbf{z}) \approx \mathbf{x}$. All diffusion training and generation is performed in this lower-dimensional latent space to reduce memory and computational.

## 4 Method

We propose LuxDiT, a diffusion-based generative framework for estimating high-dynamic-range (HDR) environment maps from a single image or video. We tailor a recent video diffusion transformer architecture [63] for lighting estimation, by jointly processing denoising targets (environment lighting) and condition tokens (LDR input images) through self-attention layers. Since a single image can be treated as a one-frame video, we refer to both inputs uniformly as input video in the remainder of this section. An overview of the architecture is shown in Figure 2. In the following sections, we describe the model design, data sources, and training procedure.

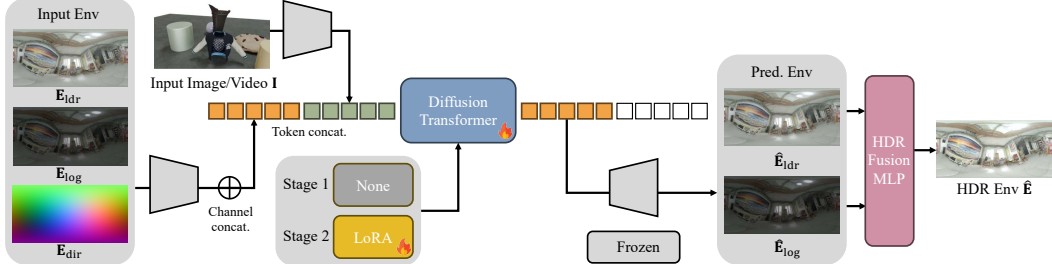

Figure 2: **Method Overview.** Given an input image or video **I**, LuxDiT predicts an environment map **E** as two tone-mapped representations, $\mathbf{E}_{\text{ldr}}$ and $\mathbf{E}_{\text{log}}$, guided by a directional map $\mathbf{E}_{\text{dir}}$. Environment maps are encoded with a VAE, and the resulting latents are concatenated and jointly processed with visual input by a DiT. The outputs $\mathbf{E}_{\text{ldr}}$ and $\mathbf{E}_{\text{log}}$ are decoded and fused by a lightweight MLP to reconstruct the final HDR panorama.

## 4.1 Model Design

We formulate HDR environment map estimation as a conditional denoising task. Given an input video $\mathbf{I} \in \mathbb{R}^{L \times H \times W \times 3}$ with $L$ frames, the model generates a corresponding sequence of $360°$ HDR panoramas $\mathbf{E} \in \mathbb{R}^{L \times H_{\text{e}} \times W_{\text{e}} \times 3}$.

Two core challenges arise: (1) standard VAEs used in latent diffusion models are trained on LDR images and cannot faithfully encode HDR content, and (2) the output panoramas are not spatially aligned with the input, requiring flexible conditioning mechanisms. We address these challenges using a dual-tonemapping HDR representation, token-based conditioning, and a unified transformer architecture that jointly denoises two latent representations of lighting.

**HDR lighting representation.** Realistic lighting involves high-intensity components such as the sun or artificial sources, with radiance values often exceeding 100 or 1,000. Representing this range in latent space is non-trivial: standard VAEs are trained on $[0, 1]$-normalized LDR images and cannot reconstruct such dynamic content, and retraining on HDR data is impractical due to data scarcity

Inspired by prior works [27, 35], we represent each HDR panorama **E** using two complementary tonemapped representations:

$$\mathbf{E}_{\text{ldr}} = \frac{\mathbf{E}}{1 + \mathbf{E}} \cdot \left(1 + \frac{\mathbf{E}}{M_{\text{ldr}}^2}\right); \qquad \mathbf{E}_{\text{log}} = \frac{\log(1 + \mathbf{E})}{\log(1 + M_{\text{log}})} \tag{2}$$

where $\mathbf{E}_{\text{ldr}}$ is a standard Reinhard tonemapping and $\mathbf{E}_{\text{log}}$ captures normalized log-intensity. We set $M_{\text{ldr}} = 16$ and $M_{\text{log}} = 10{,}000$. Both outputs are clipped to $[0, 1]$ before VAE encoding.

At inference time, the HDR environment map is reconstructed using a lightweight MLP $\psi$:

$$\hat{\mathbf{E}} = \psi\left(\mathbf{E}_{\text{ldr}}, \mathbf{E}_{\text{log}}\right). \tag{3}$$

**Diffusion latents.** Our model builds on a transformer-based diffusion model $\boldsymbol{\mu}_\theta$, adapted to predict HDR environment maps from visual input. The model operates in latent space and jointly denoises two tonemapped representations of the HDR lighting.

The tonemapped inputs $\mathbf{E}_{\text{ldr}}$ and $\mathbf{E}_{\text{log}}$ are encoded by the pretrained VAE into latent tensors $[\mathbf{z}^{\text{ldr}}, \mathbf{z}^{\text{log}}]$ with shape as $\mathbb{R}^{l \times h_{\text{e}} \times w_{\text{e}} \times C}$. These are concatenated along the channel dimension to form the diffusion target $\mathbf{z} = [\mathbf{z}^{\text{ldr}}, \mathbf{z}^{\text{log}}] \in \mathbb{R}^{l \times h_{\text{e}} \times w_{\text{e}} \times 2C}$. The input and output projection layers of the diffusion network $\boldsymbol{\mu}_\theta$ are extended to accommodate the increased channel dimension.

**Conditioning visual input in DiT.** Accurate lighting estimation requires the model to extract fine-grained shading cues from the input image, such as shadow orientation, surface reflections, and specular highlights. Unlike pixel-aligned image-to-image translation tasks, we empirically observe that concatenating conditions to the noisy latents leads to poor performance (see Table 7), indicating the need for a more flexible conditioning mechanism.

To this end, we adopt a fully attention-based architecture for the input video conditions. Specifically, we encode the input video $\mathbf{I} \in \mathbb{R}^{L \times H \times W \times 3}$ into a latent tensor $\mathcal{E}(\mathbf{I}) \in \mathbb{R}^{l \times h \times w \times C}$ using the pretrained VAE encoder, and flatten it into a token sequence $\mathbf{c} \in \mathbb{R}^{lhw \times C}$. To help the model distinguish between condition tokens and denoising targets, we apply separate adaptive layer normalization (AdaLN) [44, 63] modules to each token type at every transformer block.

**Directional embedding.** To improve angular continuity in the predicted panoramas, we inject directional information into the model. Specifically, we construct a direction map of unit vectors $\mathbf{E}_{\text{dir}}$ that encodes per-pixel lighting directions in the camera coordinate system. This map is passed through the same VAE encoder $\mathcal{E}$, then projected and fused into the noise tokens using channel-wise concatenation before the transformer blocks. During training, we apply random horizontal rotations to $\mathbf{E}_{\text{dir}}$ to encourage rotational equi-variance and robust directional encoding.

**Conditioned denoising process.** To put it together, at each denoising timestep $t$, the model receives a noisy latent $\mathbf{z}_t = [\mathbf{z}_t^{\text{ldr}}, \mathbf{z}_t^{\text{log}}]$ and predicts the corresponding clean latents conditioned on visual input as $\boldsymbol{\mu}_\theta(\mathbf{z}_t; \mathbf{c}, t)$. This transformer-based design allows the model to propagate indirect lighting cues—such as shadows and reflections—through global self-attention, enabling lighting prediction that is both scene-consistent and directionally accurate.

## 4.2 Data Strategy

Supervised training of our model requires paired data in the form $(\mathbf{I}, \mathbf{E}_{\text{ldr}}, \mathbf{E}_{\text{log}})$, where $\mathbf{I}$ is an LDR input and $\mathbf{E}_{\text{ldr}}, \mathbf{E}_{\text{log}}$ are tonemapped versions of the target HDR environment map. To overcome the scarcity of real-world HDR annotations, we leverage three complementary data sources: synthetic renderings, HDR panorama images, and LDR panoramic videos.

**Synthetic rendering data.** To supervise lighting prediction using physically accurate visual cues, we generate synthetic data by rendering randomized 3D scenes lit by HDR environment maps. Each scene consists of (i) a ground plane with randomly assigned PBR materials, (ii) 3D objects sampled from Objaverse [10], and (iii) simple geometric primitives such as spheres, cubes, and cylinders with varied materials. We render multiple frames per scene with randomized camera trajectories and environment map rotations. Despite their simplicity, these scenes exhibit diverse lighting effects, including cast shadows, specular highlights, and inter-reflections, all paired with ground-truth HDR illumination. Empirically, we find this data is critical for enabling the model to learn accurate shading cues and light-source location (see Table 7).

**HDR panorama images.** We generate training pairs by sampling perspective crops from HDR environment maps with data augmentation. Specifically, given a panorama, we randomly sample camera parameters including azimuth, elevation, field of view, and exposure scale. These parameters define a virtual pinhole camera, which we use to project the panorama into an LDR perspective view $\mathbf{I}$. The corresponding HDR environment map serves as the ground truth lighting target $\mathbf{E}$. To support temporal training, we extend this procedure to generate multi-frame sequences by smoothly varying the camera pose over time.

**LDR panorama videos.** To enable the generation of dynamic panorama environment maps, we also incorporate training data from LDR panoramic videos. Although ground-truth HDR environment maps are not available for this source, we use it in the form $(\mathbf{I}, \mathbf{E}_{\text{ldr}}, \varnothing)$, where $\mathbf{E}_{\text{ldr}}$ is derived using tonemapping and $\varnothing$ indicates the absence of log-space intensity. The panoramic video is projected into a perspective-view video using randomized camera parameters, following the same procedure as above. Despite the lack of HDR intensity, this data improves robustness and temporal consistency by exposing the model to natural image statistics, motion patterns, and diverse real-world lighting conditions. We use 2,000 panoramic videos from the WEB360 dataset [57] for training, and hold out 114 videos for evaluation.

## 4.3 Training Scheme

We adopt a two-stage training strategy to progressively build the model's capacity and improve generalization. The first stage focuses on learning physically grounded lighting cues from synthetic data. The second stage adapts the model to real-world distributions through LoRA-based fine-tuning.

**Stage I: Synthetic supervised training.** We begin by training the model on the synthetic rendering dataset described in Section 4.2. This stage enables the model to learn the fundamental relationship between image-based shading cues and HDR environment lighting.

We follow the standard DDPM training objective [21] adopted by the CogVideoX base model [63]:

$$\mathcal{L}_{\text{I}}(\theta) = \mathbb{E}_{\mathbf{z}_0, \epsilon \sim \mathcal{N}(\mathbf{0}, \mathbf{I}), t \sim \mathcal{U}(T)} \left[ \| \epsilon - \boldsymbol{\mu}_\theta(\mathbf{z}_t, \mathbf{c}, t) \|_2^2 \right], \tag{4}$$

where $\mathbf{z}_0$ denotes the clean latent pair $[\mathbf{z}^{\text{ldr}}, \mathbf{z}^{\text{log}}]$, and $\mathbf{c}$ is the conditioning latent from the input video. During training, we randomly drop either $\mathbf{z}^{\text{ldr}}$ or $\mathbf{z}^{\text{log}}$ with probability $p = 0.1$ to encourage robustness to missing tonemapped representations.

Table 1: Comparison of our method with baselines on three benchmark datasets. The results are reported in terms of scale-invariant RMSE, angular error, and normalized RMSE.

| Dataset | Method | Scale-invariant RMSE ↓ | | | Angular Error ↓ | | | Normalized RMSE ↓ | | |
|---------|--------|---------|-------|--------|---------|-------|--------|---------|-------|--------|
| | | Diffuse | Matte | Mirror | Diffuse | Matte | Mirror | Diffuse | Matte | Mirror |
| Laval Indoor | StyleLight | 0.135 | 0.315 | **0.552** | 4.238 | 4.742 | 6.781 | 0.234 | 0.404 | 0.511 |
| | DiffusionLight | 0.124 | 0.325 | 0.597 | **2.500** | **3.421** | 5.936 | 0.216 | 0.361 | **0.431** |
| | Ours | **0.112** | **0.297** | 0.586 | 2.555 | 3.526 | **5.641** | **0.196** | **0.341** | 0.457 |
| Laval Outdoor | H-G et al. [23] | 0.300 | 0.437 | 0.587 | 7.851 | 8.755 | 26.052 | 0.551 | 0.627 | 0.740 |
| | NLFE | 0.112 | 0.234 | 0.431 | 4.804 | 5.279 | 7.278 | 0.217 | 0.331 | 0.496 |
| | DiffusionLight | 0.083 | 0.224 | 0.414 | **1.936** | 2.955 | 5.491 | 0.167 | 0.330 | 0.472 |
| | Ours | **0.068** | **0.190** | **0.396** | 2.018 | **2.939** | **5.286** | **0.137** | **0.271** | **0.454** |
| Poly Haven | StyleLight | 0.138 | 0.336 | 0.620 | 3.034 | 4.272 | 6.602 | 0.198 | 0.344 | 0.474 |
| | NLFE | 0.159 | 0.326 | 0.571 | 3.305 | 4.240 | 5.180 | 0.224 | 0.365 | 0.458 |
| | DiffusionLight | 0.113 | 0.270 | 0.519 | 2.199 | 3.121 | 4.104 | 0.191 | 0.282 | 0.391 |
| | Ours | **0.077** | **0.196** | **0.442** | **1.235** | **1.977** | **2.783** | **0.111** | **0.199** | **0.323** |

Table 2: Angular error on estimated peak luminance light direction on Laval Outdoor sunny scenes.

| Method | Peak Angular Error ↓ | |
|--------|------|--------|
| | Mean | Median |
| H-G et al. [23] | 52.8 | 47.8 |
| NLFE | 52.9 | 43.5 |
| DiffusionLight | 44.4 | 32.1 |
| Ours | **23.7** | **17.5** |

Table 3: Quantitative comparison with video input. Peak angular error (PAE) is used to evaluate PolyHaven-Peak videos. Angular error (AE) on is used to evaluate WEB360 LDR videos.

| Method | PolyHaven-Peak | | WEB360 | |
|--------|----------|---------|------|--------|
| | PAE Mean ↓ | PAE Std ↓ | AE ↓ | AE Std ↓ |
| DiffusionLight | 19.09 | 10.31 | 6.504 | 0.269 |
| Ours (image) | 5.74 | 3.68 | 5.679 | 0.382 |
| Ours (video) | **5.21** | **1.95** | **5.218** | **0.072** |

**Stage II: Semantic adaptation.** After base training, we fine-tune the model to improve semantic alignment between the input appearance and the predicted HDR environment map.

This stage uses real-world data sources, including perspective projections from HDR panoramas and LDR panoramic videos. Since HDR ground truth is not available in the latter, we supervise only the LDR-tonemapped component. To avoid overfitting and preserve the pretrained model capacity, we apply parameter-efficient LoRA fine-tuning [24], optimizing a small set of injected low-rank parameters $\Delta\theta$ in the transformer layers:

$$\mathcal{L}_{\text{II}}(\Delta\theta) = \mathbb{E}_{\mathbf{z}_0, \epsilon \sim \mathcal{N}(\mathbf{0}, \mathbf{I}), t \sim \mathcal{U}(T)} \left[ \|\epsilon - \boldsymbol{\mu}_{\theta + \Delta\theta}(\mathbf{z}_t, \mathbf{c}, t)\|_2^2 \right], \tag{5}$$

## 5  Experiments

### 5.1  Experiment Settings

**Implementation details.** We use the pre-trained CogVideoX [63] model as our backbone. All training is conducted on 16 NVIDIA A100 GPUs. Input resolutions are randomly sampled between $512 \times 512$ and $480 \times 720$, and output environment map resolutions are between $128 \times 256$ and $256 \times 512$. The image-based model is trained with a batch size of 192 for 12,000 iterations. For video training, we use the same spatial resolutions and uniformly sample frame lengths from $9, 17, 25$. The video model is trained with an average batch size of 48 for an additional 12,000 iterations. LoRA modules are applied to all attention layers with a rank of 64. We fine-tune the LoRA parameters for 5,000 iterations during the adaptation stage. Please refer to supplement for implementation details.

**Datasets.** We evaluate our method on the following three benchmark datasets, covering various indoor and outdoor scenes. 1) Laval Indoor [15]: We use the same set of 289 test HDRIs used by prior works [45, 55]; 2) Laval Outdoor [23]: We evaluate on 116 sunny HDR panoramas with concentrated sunlight selected from the original dataset; 3) Poly Haven [67]: We select 181 Poly Haven HDRIs not used during model training to evaluate performance across both indoor and outdoor scenes.

**Metrics.** Following prior works [55, 45], we use three standard metrics for evaluating HDR lighting: scale-invariant root mean square error (si-RMSE) [17], angular error in degrees [32], and normalized RMSE (n-RMSE) [45]. For scenes with concentrated sunlight, we additionally report peak angular error (PAE) [23, 58], which measures the angular deviation of the predicted peak light direction.

**Baselines.** For indoor scenes, we compare against DiffusionLight [45], StyleLight [55], Weber et al. [61], and EMLight [69], using metrics reported by [45] when applicable. For outdoor scenes, we compare against DiffusionLight [45], Hold-Geoffroy *et al.* [23], and NLFE [58].

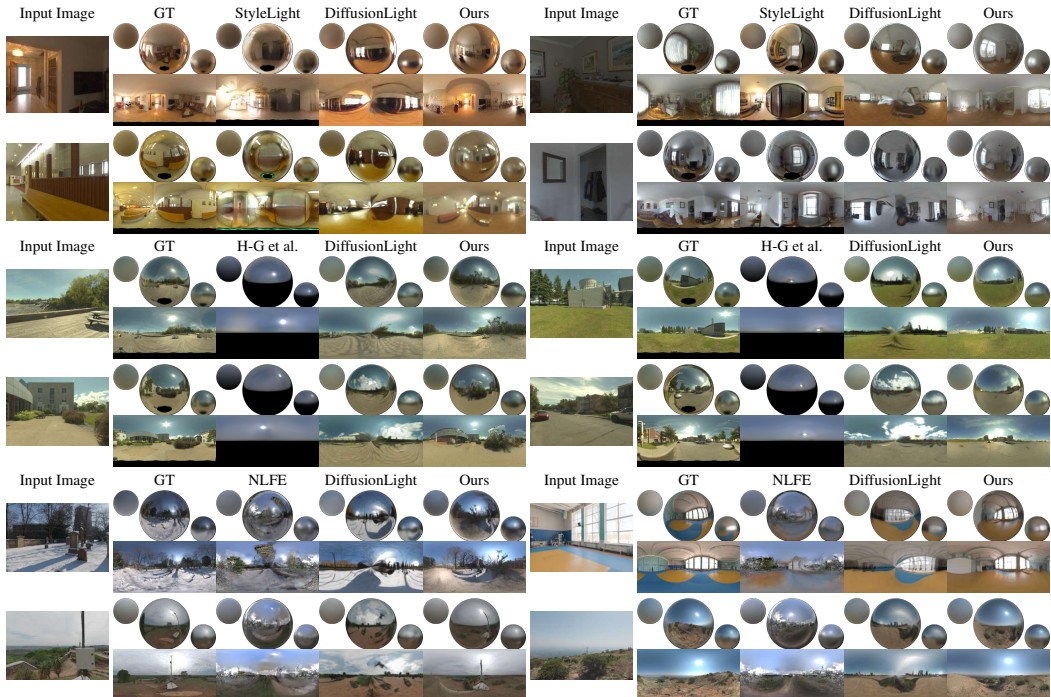

Figure 3: Qualitative comparison with baseline methods on three benchmark datasets.

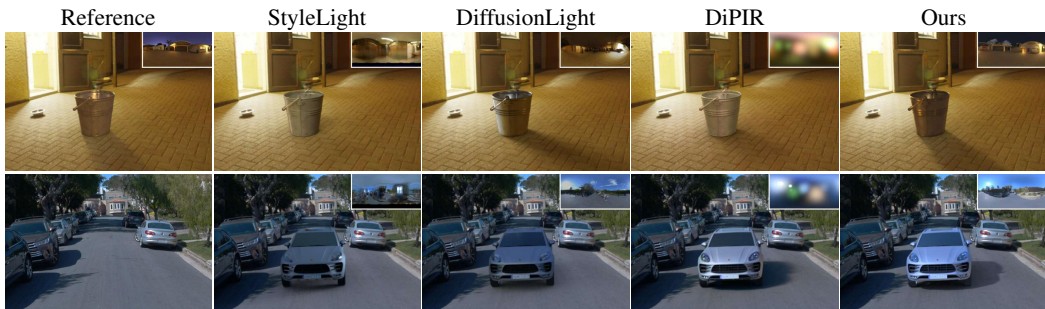

Figure 4: Qualitative comparison of virtual object insertion.

Table 4: Ablation study on impact of LoRA scale at inference time.

| LoRA Scale | Diffuse ↓ | Matte ↓ | Mirror ↓ |
|---|---|---|---|
| 0.00 | 2.98 | 5.02 | 6.07 |
| 0.25 | 2.09 | 3.69 | 4.67 |
| 0.50 | 1.52 | 2.66 | 3.56 |
| 0.75 | 1.22 | 2.05 | 2.88 |
| 1.00 | **1.17** | **1.92** | **2.72** |

Table 5: Ablation study on impact of camera field-of-view.

| FOV | Diffuse ↓ | Matte ↓ | Mirror ↓ |
|---|---|---|---|
| 45° | 1.29 | 2.14 | 2.95 |
| 50° | 1.26 | 2.06 | 2.86 |
| 60° | 1.17 | 1.92 | 2.72 |
| 70° | 1.15 | 1.85 | 2.63 |
| 75° | **1.13** | **1.80** | **2.59** |

Table 6: Ablation study on impact of camera elevation.

| Elevation | Diffuse ↓ | Matte ↓ | Mirror ↓ |
|---|---|---|---|
| −30° | 1.70 | 3.04 | 3.95 |
| −15° | 1.22 | 2.05 | 2.87 |
| 0° | **1.17** | **1.92** | **2.72** |
| 15° | 1.28 | 2.09 | 2.94 |
| 30° | 1.71 | 2.59 | 3.51 |

Table 7: Ablation study on model design choices and training data. We report the angular error with three-spheres protocol.

| Settings | Laval Indoor | | | Poly Haven | | |
|---|---|---|---|---|---|---|
| | Diffuse ↓ | Matte ↓ | Mirror ↓ | Diffuse ↓ | Matte ↓ | Mirror ↓ |
| Ours (channel concat.) | 7.09 | 10.04 | 11.07 | 7.09 | 10.04 | 11.07 |
| Ours (w/o synthetic data) | 4.50 | 5.14 | 6.96 | 1.48 | 2.08 | 2.86 |
| Ours | **2.56** | **3.53** | **5.64** | **1.23** | **1.98** | **2.78** |

## 5.2 Evaluation of Image Lighting Estimation

We follow the evaluation protocol from prior work to render spheres with three representative materials (gray-diffuse, silver-matte, and mirror), using the estimated HDR environment map from the LDR input image [15, 55, 45].

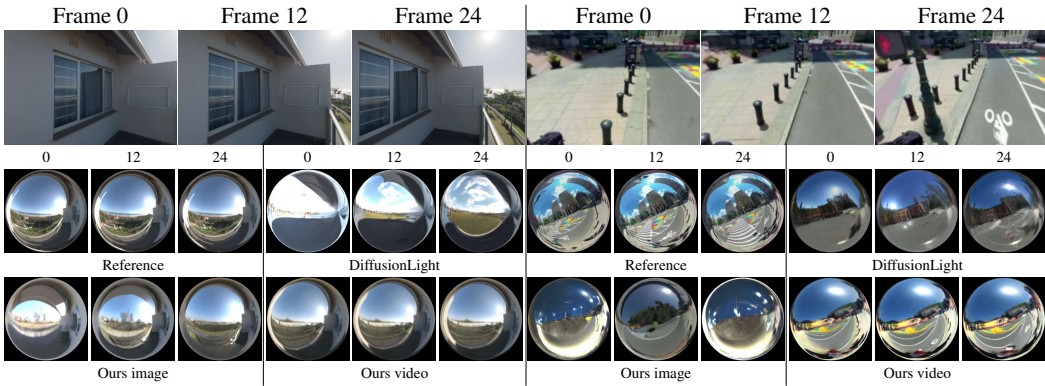

Figure 5: Qualitative comparison of video lighting estimation.

Table 1 reports quantitative comparisons on three benchmarks spanning both indoor and outdoor scenes. On the Laval Indoor dataset, our method performs comparably or better than DiffusionLight across most metrics, despite not using Laval Indoor dataset during training. This dataset exhibits a noticeable shift in color and intensity distribution compared to our training set, and our strong performance demonstrates robust generalization.

From qualitative comparison shown in Figure 3, DiffusionLight can lose angular high-frequency details from the input image due to its distorted representation. In contrast, our estimated environment maps can recover more high-frequency details while preserving accurate lighting.

On the Laval Outdoor and Poly Haven datasets with a broader dynamic range, our method consistently outperforms prior state-of-the-art methods. Hold-Geoffroy *et al.* [23] can estimate concentrated peak light source such as sunlight; however, its results do not adapt well to the details of the input image. NLFE [58] can estimate in-context environment maps, but it often fails to estimate accurate highlights. DiffusionLight performs better than other baselines, but due to its limited dynamic range, it struggles with outdoor high-intensity light sources.

To further assess directional accuracy, we evaluate the angular error of the peak luminance direction on a subset of the Laval Outdoor dataset containing direct sunlight. Table 2 reports the mean and median peak angular errors. Our method reduces peak angular error by nearly 50% compared to DiffusionLight, confirming its advantage in capturing accurate light direction—a critical factor for casting realistic shadows in downstream applications such as object insertion.

## 5.3 Evaluation of Video Lighting Estimation

To evaluate lighting estimation accuracy and consistency on video input, we construct two types of test sequences:

- *PolyHaven-Peak*: We project 12 unseen Poly Haven panoramas (each with direct sunlight) into videos using a smooth panning camera. This setting is used to evaluate peak angular error.
- *WEB360*: We randomly select 12 LDR panoramic videos featuring dynamic content from WEB360 and render them into perspective views with fixed horizontal camera motion. This setting evaluates temporal consistency using chromatic angular error on rendered mirror spheres.

Each set contains 12 videos at resolution of $480 \times 720$ and a length of 25 frames. To quantify temporal consistency, we compute the standard deviation (std) of per-frame error metrics for each video clip, and average the results across the 12-video set.

We compare our video inference to two baselines: our own image-based inference (applied frame-by-frame) and DiffusionLight [45]. Table 3 reports the results. Our method outperforms DiffusionLight. Comparing to Ours (image), video inference achieves higher accuracy and significantly lower temporal variance, indicating more stable predictions across time.

Figure 5 shows qualitative examples of video inference. Both DiffusionLight and our image-based variant exhibit visible temporal flickering. In contrast, our method produces smooth lighting transitions, successfully aligning content across frames and preserving consistent lighting behavior over time.

## 5.4 Evaluation of Virtual Object Insertion

Virtual object insertion is a key downstream application of lighting estimation. We evaluate our method on this task using the benchmark from [36], using 11 HDR panoramas from the Poly Haven dataset [67]. For each scene, a virtual object and a known ground plane are manually placed into the environment. Each test case includes an LDR background image rendered from the HDR panorama, along with a posed object and ground plane. A pseudo-ground-truth object insertion is generated by rendering the object using the original HDR environment map. This allows for controlled comparison against renderings produced using predicted lighting.

Table 8: Quantitative evaluation of virtual object insertion. We report the percentage of images where users preferred Ours over baselines. A preference $> 50\%$ indicates Ours outperforming baselines.

| Method | RMSE ↓ | SSIM ↑ | Ours Preferred |
|---|---|---|---|
| StyleLight | 0.056 | 0.986 | 60.6% |
| DiffusionLight | 0.057 | 0.987 | 60.6% |
| DiPIR | 0.048 | 0.989 | 54.5% |
| Ours | 0.047 | 0.990 | / |

We report quantitative metrics in Table 8. In addition, we conduct a user study to assess perceptual quality (details provided in the supplement), and report the percentage of samples where users preferred our results over baseline methods.

Our method achieves visual quality comparable to DiPIR and significantly outperforms other baselines. Notably, DiPIR is specialized for object insertion and incorporates additional modules for tone mapping and appearance harmonization. In contrast, our model estimates lighting alone, yet still produces realistic composite renderings. We include qualitative results in Figure 4.

## 5.5 Ablation Study

**Model Design and Training Data** We evaluate two model variants to ablate the contributions of our architectural and training design: (1) *Channel concatenation*: This variant fuses input and environment map (resized to match input image) latents along the channel dimension [27], and no token-wise concatenation is used. Our two-stage training is also applied. (2) *Training without synthetic data*: This variant skips Stage I training and uses only panorama crops for fine-tuning.

Table 7 reports angular errors on Laval Indoor and Poly Haven. Channel concatenation significantly underperforms, confirming the importance of token-level conditioning. Without synthetic pretraining, the model performs well in-domain (Poly Haven) but degrades out-of-domain (Laval Indoor), showing synthetic data pre-training is crucial for learning generalized lighting priors.

**LoRA scale.** We vary the LoRA interpolation weight from 0.0 to 1.0 to ablate how fine-tuned LoRA affects the predicted lighting content. Table 4 shows that higher LoRA weights yield lower angular error on Poly Haven, validating the effectiveness of LoRA for improving semantic alignment.

**Camera sensitivity.** We test robustness to camera variation by rendering crops from Poly Haven under varying field of view ($45°$ to $75°$) and camera elevation ($-30°$ to $30°$). Results in Tables 5 and 6 show that while extreme viewpoints introduce mild error increases, performance remains stable, demonstrating robustness to moderate viewpoint shifts.

## 6 Discussion

We introduce LuxDiT, a conditional generative model for estimating HDR scene illumination from casually captured images and videos. Our approach fine-tunes a video diffusion transformer (DiT) to synthesize HDR environment maps, combining large-scale synthetic data for learning physically grounded priors with LoRA-based adaptation on real HDR panoramas to improve semantic alignment. Extensive experiments demonstrate that LuxDiT produces accurate, high-frequency, and scene-consistent lighting predictions from limited visual input.

**Limitations and future work.** While LuxDiT produces high-quality lighting predictions, inference remains computationally intensive due to the iterative nature of diffusion models, limiting its use in real-time applications. Future work could explore model distillation or more efficient architectures to accelerate inference. Additionally, the resolution of predicted panoramas is limited by data and training scale; generating high-resolution outputs for immersive applications will require richer, more diverse HDR supervision. Looking ahead, with recent progress in joint generative modeling [5, 37], we see LuxDiT as a step toward unified inverse and forward rendering frameworks, complementing recent progress in neural forward rendering and G-buffer estimation [68, 35, 20]. Future directions include joint modeling or co-training of lighting, geometry, and material for general-purpose scene reconstruction and appearance synthesis.

## Acknowledgments and Disclosure of Funding

The authors thank Jacob Munkberg, Jon Hasselgren, Jun Gao, and Huan Ling for their insightful discussions. This work was conducted at and supported by NVIDIA.

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

# Appendix

In the supplementary material, we discuss the broader impact of our project in Sec. A, and provide additional details for implementation and experiments in Sec. B. Sec. C provides additional quantitative and qualitative results. We refer to the accompanied video for extended comparisons on video lighting estimation.

## A  Broader Impact

We introduce LuxDiT, a generative model for estimating high-dynamic-range (HDR) environment lighting from casually captured images and videos. Lighting estimation is a core challenge in photorealistic rendering due to its non-local and indirect nature. LuxDiT produces scene-consistent HDR panoramas, enabling applications in virtual object insertion, relighting, AR/VR, and visual effects. It can also support synthetic data generation for downstream tasks in robotics and perception, where realistic illumination is critical.

Similar to other generative methods, LuxDiT could be misused to produce visually convincing but deceptive content. While it does not directly generate synthetic scenes, it enables realistic virtual object insertion and may facilitate the creation of manipulated imagery that is difficult to distinguish from real footage. We encourage responsible use of LuxDiT and caution against its deployment in contexts where synthetic content could mislead viewers or undermine public trust, such as misinformation or falsified media.

## B  Additional Details

### B.1  HDR Reconstruction

Section 4.1 describes our method for reconstructing HDR environment maps from two tone-mapped LDR images using a lightweight MLP $\psi\left(\mathbf{E}_{\text{ldr}}, \mathbf{E}_{\text{log}}\right)$. This MLP consists of 5 layers with 64 hidden units per layer and LeakyReLU activation. A softplus activation is applied to the final output layer to ensure non-negative outputs.

The MLP $\psi$ operates on a per-pixel basis: it takes a pair of LDR RGB values as input and predicts a single HDR RGB value. It is trained using the same HDR environment maps as the diffusion model, with augmentations including random intensity rescaling and exposure adjustments for diversity. To simulate limited input precision, LDR inputs are randomly quantized to 8-bit RGB values. We train the MLP using a Huber loss with $\delta = 1.0$, which provides robustness against large HDR outliers while preserving smooth gradients.

Additionally, we show the tone-mapping curves used to generate the LDR images in Fig. 6. Our dual-tone mapping strategy ensures sufficient sampling across the full dynamic range $[0, 10{,}000]$, supporting accurate HDR reconstruction.

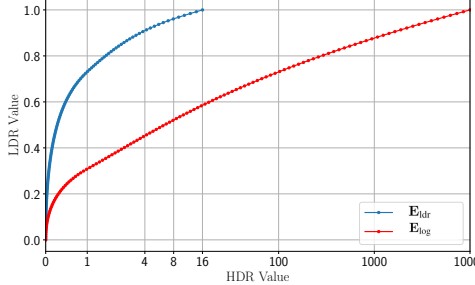

Figure 6: The two tone-mapping curves used to generate the LDR images. The 128 dot points along the curve are evenly spaced along $[0, 1]$ LDR value range.

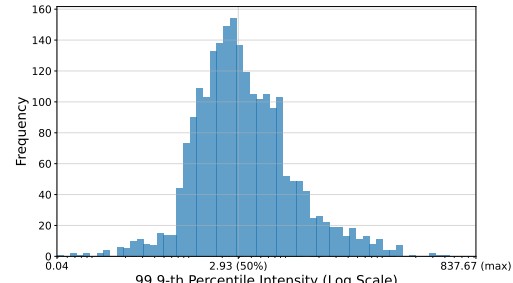

Figure 7: The histogram of the 99.9-th percentile intensity of all HDR environment maps in our training set.

## B.2 Datasets

We provide more details about the datasets used in our experiments.

**The data sources of HDR environment maps.** We collected 2386 HDR environment maps from the following 4 data sources either publicly available or commercially available.

- *Poly Haven*[0]: 626 HDR environment maps with a wide range of indoor and outdoor lighting.
- *HDR Maps*[1]: 403 HDR environment maps with diverse lighting conditions, including 294 panorama maps and 109 hemi-sphere sky maps.
- *HDRI Skies*[2]: 457 HDR environment maps with outdoor lighting conditions.
- *DOSCH DESIGN*[3]: 900 HDR environment maps mainly for outdoor lighting conditions.

Figure 7 shows the histogram of the 99.9-th percentile intensity of all HDR environment maps in our training set. With over 50% of the HDR environment maps having a 99.9-th percentile intensity greater than 2.93. Note that the for outdoor lighting, the highest intensity can be orders of magnitude higher than the 99.9-th percentile. Among these, Poly Haven and HDR Maps offer greater diversity across scene types. To balance the training distribution across data sources, we apply sampling weights in the ratio $3:2:2:1$ in the order listed above.

For quantitative and qualitative evaluation, we use the Laval Indoor [4] and Laval Outdoor [5] datasets, which contain calibrated HDR panoramas of real-world indoor and outdoor scenes.

**Synthetic rendering data.** Similar to OBJect [40] and DiffusionRenderer [35], we create synthetic 3D scenes by compositing multiple 3D objects from Objaverse [10] and randomly placing them on a plane with varying plane textures. We use a filtered subset of Objaverse, containing ∼269,000 3D objects with decent geometries and material textures, to create synthetic 3D scenes. The varying plane textures are sampled from ∼4000 PBR textures from Mat-Synth[6] [54]. Each composited scene contains up to 3 sampled Objaverse objects. We additionally add up to 3 random geometry primitives (sphere, cube, and cylinder) with varying material textures to provide rich shading cues for model to learn. For each scene, we randomly render 1∼4 video clips with varying camera motions (e.g., orbiting camera and oscilating

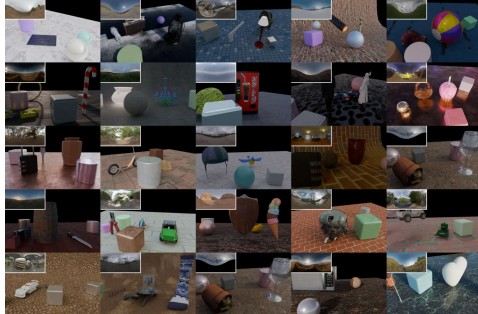

Figure 8: Randomly sampled example images from our synthetic rendering data.

camera) and environment lightings. We use a path-tracing renderer with 128 samples per pixel (spp) and the default OptiX denoiser to render the video clips with a resolution of $480 \times 720$ or $512 \times 512$. The HDR rendering results are tone-mapped to LDR images using Blender's AgX tonemapping[7]. In total, we created ∼190,000 random synthetic scenes, resulting in ∼260,000 video clips with at least 16 frames per video clip.

**Perspective crops of HDR panorama images.** We use a subset of 1251 HDR panoramas with meaningful contents from Poly Haven, HDR Maps, and HDRI Skies for the training with perspective crops. Instead of pre-processing the perspective crops from the HDR panoramas, we do the perspective crops on-the-fly during the training. The projection camera's azimuth angle is randomly sampled from $[0, 360°]$ and the elevation angle is randomly sampled from $-10°$ to $10°$. The camera's field of view (FOV) is randomly sampled from $45°$ to $80°$. The perspective crops are rendered with a resolution of $480 \times 720$. A random tone-mapping function is applied to perspective projection crops

---

[0]https://polyhaven.com/
[1]https://hdrmaps.com/
[2]https://hdri-skies.com/
[3]https://doschdesign.com/
[4]http://hdrdb.com/indoor/
[5]http://hdrdb.com/outdoor/
[6]https://huggingface.co/datasets/gvecchio/MatSynth
[7]https://www.blender.org/

to generate LDR images. The tone-mappings include ACES, Filmic, AgX, and Gamma-2.4 sRGB mappings. Auto-exposure (i.e., remapping the 99-th percentile intensity to 0.9) is also randomly applied to the LDR crops. For video input, we create trajectories of projection cameras by smoothly rotating the camera angle within an angular cone of $15°$.

**Perspective crops of LDR panorama videos.** Similar to the perspective crops of HDR panorama images. We on-the-fly sample perspective crops from the LDR panorama videos. Due to the lack of HDR content, we only apply a random auto-exposure tone-mapping to the perspective crops.

### B.3    Model Details and Initialization

LuxDiT is fine-tuned from the pre-trained CogVideoX-5b-I2V[8]. To adapt this model for our task, we replace the original text token with an image input token. This image token is generated in the same manner as the environment map noise token, but without adding noise. We reuse the model's existing text-processing layers (e.g., AdaLN) to process these new image input tokens. Furthermore, we extend the input projection layer to incorporate additional conditioning channels derived from the concatenated noise token; these extended channels are initialized to zero. Similarly, the output projection layer is extended to predict dual tone-mapped environment tokens, with its newly added channels initialized from the original model's weights.

### B.4    User Study Details for Virtual Object Insertion

Following prior works [14, 16, 15, 58, 36], we conduct a user study on Amazon Mechanical Turk to compare our method against baseline approaches in terms of perceptual realism for virtual object insertion. Each participant is shown a pair of rendered results—one from our method and one from a baseline—and asked to assess lighting realism, focusing on shadows, reflections, and overall visual integration.

The specific instructions shown to participants are:

> Instruction: Find the inserted virtual object, look at the difference, and select the more realistic image.
>
> An AI system is trying to insert a virtual object into an image in a natural way. It aims to make the virtual object look as if it is part of the scene. There are two results: Trial A and Trial B, and the virtual object is located in the center of each image. Please zoom in to compare the differences between the two images, and pay attention to the lighting effects such as the reflections and shadows.
>
> Which one looks more realistic?
> ☐ A
> ☐ B

Participants are required to use a monitor 24 inches or larger. Image pairs are randomly shuffled to prevent bias. Following [36], we repeat the user study three times, and recruited 11 unique participants for each experiment. We compute the percentage of *images* for which users preferred our method over the baseline, and report the average user preferences for three repeated experiments. In total, the study includes $11 \times 3 \times 11 \times 3 = 1089$ individual comparisons.

### B.5    Three-sphere Evaluation Protocol

We adopt the three-sphere rendering setting described in StyleLight [55], with evaluation scripts provided by DiffusionLight[9].

For the Laval Indoor dataset, we use the same set of HDR environment maps and corresponding perspective crops as DiffusionLight. We resize and crop the input image to $480 \times 720$ for our model. For Laval Outdoor and Poly Haven environment maps, we generate perspective crops using a fixed horizontal camera with a $60°$ field of view and a resolution of $480 \times 720$. For Laval Outdoor, we apply auto-exposure by scaling the 50th percentile intensity to 0.5.

---

[8] https://huggingface.co/THUDM/CogVideoX-5b-I2V
[9] https://github.com/DiffusionLight/DiffusionLight-evaluation

## C  Additional Experiments

### C.1  Array-of-Spheres Evaluation

Following prior work [61, 8], we evaluate our method using the array-of-spheres protocol, which renders a grid of diffuse spheres on a ground plane using the predicted environment map.

We use 2,240 perspective crops from 224 Laval Indoor panoramas, provided by DiffusionLight[10]. All input images are resized to $512 \times 512$ to match our model input. Quantitative results are shown in Table 9 and qualitative results in Fig. 9.

While our method performs slightly below specialized systems like Weber *et al.* [61] and EMLight [69], it remains competitive—despite not being trained on Laval Indoor. Notably, it outperforms StyleLight [55] and DiffusionLight [45], demonstrating strong generalization across lighting domains.

Table 9: Scores on indoor array-of-spheres.

| Method | si-RMSE ↓ | AE ↓ |
|---|---|---|
| EverLight [8] | 0.091 | 6.36 |
| StyleLight [55] | 0.123 | 7.09 |
| Weber et al. [61] | **0.081** | 4.13 |
| EMLight [69] | 0.099 | **3.99** |
| DiffusionLight [45] | 0.090 | 5.25 |
| Ours | 0.089 | 4.90 |

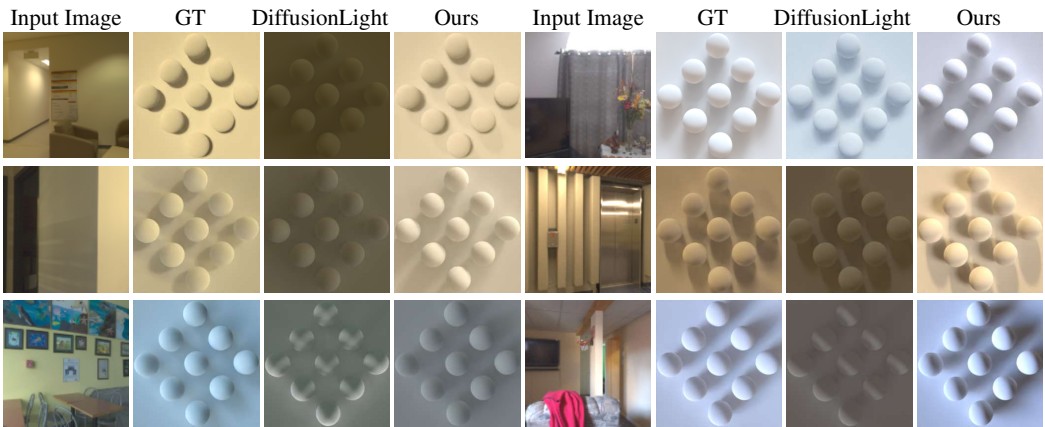

Figure 9: Visual results on array-of-spheres protocol.

### C.2  Lighting Estimation with the Cube++ Dataset

We also evaluated our method on the Cube++ dataset [12], specifically designed for illumination estimation and color constancy. This dataset includes illumination information annotated by the SpyderCube [11]. For our experiment, we selected 100 processed JPEG images from Cube++. We then applied both DiffusionLight and our method to estimate the illumination from each image. Subsequently, we rendered the left and right white faces of the SpyderCube under the estimated illumination, assuming purely Lambertian diffuse surfaces. To prevent information leakage from the SpyderCube in the input images, we masked out the SpyderCube from the tested images and inpainted the masked region using LaMa [51]. We then compared the rendered face colors to the colors sampled directly from the SpyderCube JPEG images. Table 10 presents the RMSE and angular errors, demonstrating that our method clearly outperforms DiffusionLight, achieving angular errors of less than 5° on both faces. Visual comparison results are further illustrated in Fig. 10.

Table 10: Scores on SpyderCube white face rendering on Cube++ dataset.

| Method | RMSE ↓ | | AE ↓ | |
|---|---|---|---|---|
| | Left | Right | Left | Right |
| D.Light [45] | 0.044 | 0.035 | 7.221 | 5.741 |
| Ours | **0.024** | **0.025** | **3.985** | **4.003** |

---

[10] https://github.com/DiffusionLight/image-array_of_spheres
[11] https://www.datacolor.com/spyder/products/spyder-cube/

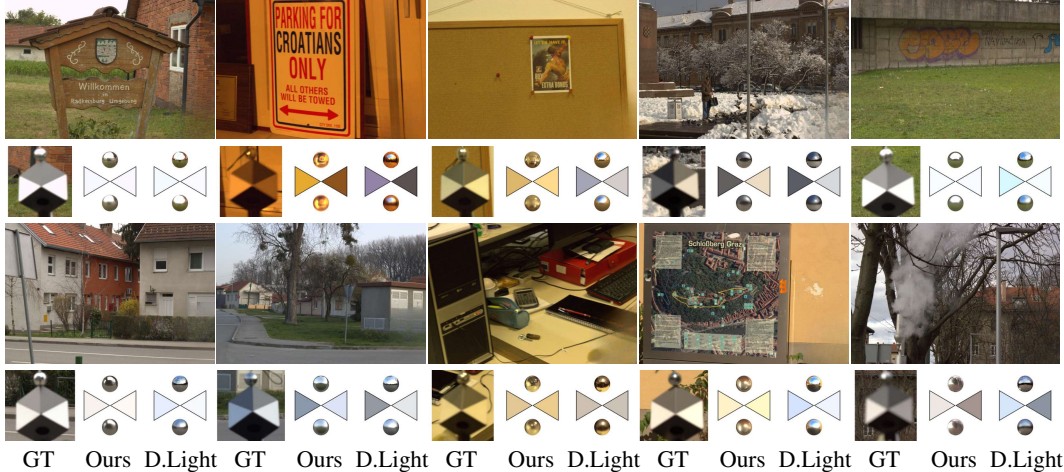

GT  Ours D.Light GT  Ours D.Light GT  Ours D.Light GT  Ours D.Light GT  Ours D.Light

Figure 10: Visual results on Cube++ dataset. We show the rendered two white cube faces, mirror ball, and matte silver ball from our method and DiffusionLight for visual comparison.

## C.3 Lighting Estimation from Foreground Objects

Since our model is trained on object-centric synthetic rendering data, we can also apply it to estimate lighting from foreground objects. We selected 4 NeRF synthetic objects [41] and 4 real-world objects [53], aiming to estimate lighting from videos containing nine consecutive rendering views.

We qualitatively compare LuxDiT with optimization-based inverse rendering methods [42, 18] that reconstruct 3D geometry and lighting from full NeRF scenes. Using the ground truth camera poses, we rotate each frame's estimated lighting into the global coordinate system and average across frames to produce the final environment map.

Qualitative results are shown in Fig.11. On mostly diffuse objects like lego and hotdog, our method recovers highlight directions accurately, enabling shadow rendering consistent with the input. For glossy objects like mic and ficus, our model estimates lighting nearly identical to the ground truth. While these HDR environment maps are included in our training set, the NeRF scenes differ significantly from our synthetic renderings (see Fig. 8), indicating that our model leverages shading cues and learned priors rather than direct memorization. In contrast, optimization-based baselines struggle to capture high-frequency lighting detail and often introduce noise and artifacts in lighting.

We further tested our method on real-world foreground objects from the Objects-with-Lighting dataset [53], which provides ground truth distant environment lighting. Similar to the NeRF synthetic scene setup, the estimated lighting was then aligned into the global coordinate system using ground truth camera poses. We compared our approach to NeuS+Mitsuba [56, 26], the top-performing method on this dataset [53]. The metrics, using the three-sphere protocol, are presented in Table 11, with visual results in Fig. 12.

While our model performs well overall, minor errors remain, *e.g.* color shifts in the NeRF Lego scene (Fig. 11) and a slightly higher si-RMSE compared to NeuS+Mitsuba (Table 11). We believe combining our generative model with optimization-based methods could further enhance lighting estimation, which we leave for future work.

Table 11: Comparison of our method with NeuS+Mitsuba on Objects with Lighting datasets.

| Method | Scale-invariant RMSE ↓ | | | Angular Error ↓ | | | Normalized RMSE ↓ | | |
|---|---|---|---|---|---|---|---|---|---|
| | Diffuse | Matte | Mirror | Diffuse | Matte | Mirror | Diffuse | Matte | Mirror |
| NeuS+Mitsuba | 0.082 | 0.232 | 0.424 | 3.145 | 3.383 | 3.526 | 0.180 | 0.545 | 0.717 |
| Ours | 0.086 | 0.253 | 0.482 | 1.262 | 1.594 | 2.000 | 0.153 | 0.339 | 0.479 |

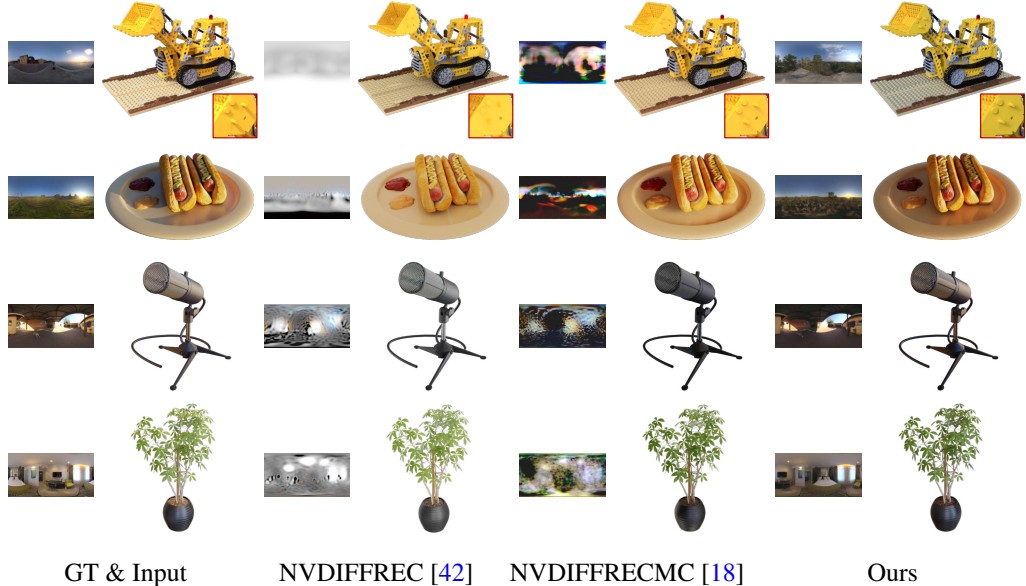

| GT & Input | NVDIFFREC [42] | NVDIFFRECMC [18] | Ours |

Figure 11: Lighting estimation from the NeRF synthetic objects. We use the estimated lighting from different methods to re-render the original NeRF Blender scenes.

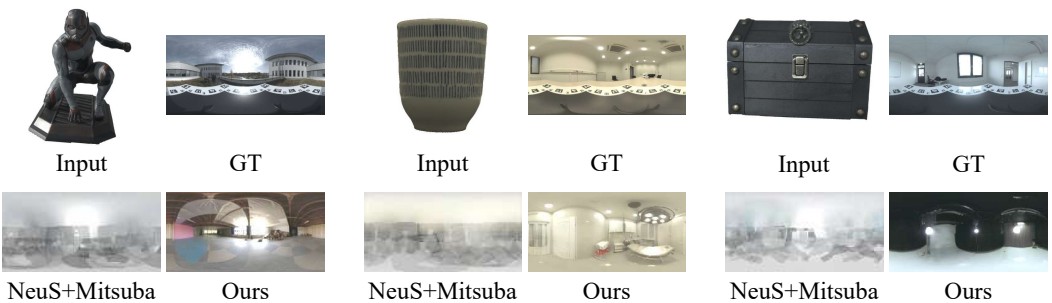

| Input | GT | Input | GT | Input | GT |

| NeuS+Mitsuba | Ours | NeuS+Mitsuba | Ours | NeuS+Mitsuba | Ours |

Figure 12: Lighting estimation from the masked real objects from Objects with Lighting.

## C.4  Additional Ablations

### C.4.1  The Choice of the HDR Fusion Model

As detailed in Sec. 4.1, a lightweight MLP $\psi$ is employed to merge the dual-tonemapped environment maps, $\mathbf{E}_{\text{ldr}}$ and $\mathbf{E}_{\text{log}}$, thereby reconstructing the HDR environment map $\hat{\mathbf{E}}$. There are also alternative fusion methods, such as using a more complex CNN model to incorporate adjacent pixel information for HDR fusion, or applying a rule-based approach with explicit inverse equations. To justify our choice of a simple MLP, we evaluate various HDR fusion techniques, including MLP, CNN, and a rule-based method. The CNN model has an identical number of layers to our MLP model, using $3 \times 3$ convolution kernels across layers. The rule-based method involves applying the inverse Reinhard map for lights with intensity below 8, a linear interpolation between Reinhard and log maps for intensities ranging from 8 to 16, and exclusively the log map for intensities exceeding 16.

Table 12 presents the RMSE results on testing Polyhaven HDRIs. All three methods demonstrate comparable accuracy, with the MLP approach exhibiting a slight advantage. Compared to the rule-based approach, we believe the neural approach can better handle numerical inconsistency after image uint8 quantization, and the potential data range overflow (e.g., lights beyond the pre-defined maximum intensity 10000).

Table 12: Comparison on different HDR fusion approaches.

|  | MLP | CNN | Rule |
|---|---|---|---|
| RMSE↓ | 11.55 | 11.74 | 11.71 |

### C.4.2 The Impact of LoRA on Synthetic Scenes

Section 5.5 demonstrates the impact of varying LoRA scales (0.0 to 1.0) on the predicted lighting content of real-world images. This ablation study, conversely, investigates how our LoRA model, trained with real images, affects the lighting estimation of synthetic foreground objects. Table 13 presents the angular errors using a three-sphere evaluation, and Fig. 16 provides the visual results.

Table 13: Ablation study on impact of LoRA scale on synthetic foreground objects.

| LoRA Scale | 0.00 | 0.25 | 0.50 | 0.75 | 1.00 |
|---|---|---|---|---|---|
| **Diffuse** ↓ | 1.594 | 1.737 | 2.170 | 3.832 | 3.937 |
| **Matte** ↓ | 2.068 | 2.311 | 2.914 | 5.322 | 5.891 |
| **Mirror** ↓ | 3.405 | 3.690 | 4.342 | 6.783 | 7.400 |

In contrast to the ablation performed on scene images, a larger LoRA scale leads to lower lighting estimation accuracy. As Fig. 16 illustrates, increasing the LoRA scale causes foreground content to gradually appear on the estimated environment map, which is consistent with our LoRA model's behavior. Nevertheless, the estimated highlights remain consistent across different LoRA scales.

## D    Additional Results

We provide additional visual results in this section to further support the claims made in the main paper.

- Model Ablation and LoRA Scale: Figure 13 details the ablation study on our model's design and the exploration of different LoRA scales.
- Camera Parameter Variations: Figures 14 and 15 show lighting estimation performance when varying camera field of view (FOV) and elevation angles, respectively.
- Three-Sphere Rendering Evaluations: Figures 17, 18, and 19 display further lighting estimation outcomes using the three-sphere rendering protocol on the Laval Indoor, Laval Outdoor, and Poly Haven datasets.
- Virtual Object Insertion: Figures 20 and 21 illustrate additional virtual object insertion results on Poly Haven panorama crops and Waymo driving scenes.

| Input Image | Model Ablation | LoRA Scale Exploration | | | |
|---|---|---|---|---|---|
| | channel concat. | LoRA 0.0 | LoRA 0.1 | LoRA 0.2 | LoRA 0.4 |
| | w/o syn. data | LoRA 0.6 | LoRA 0.7 | LoRA 0.8 | LoRA 1.0 |

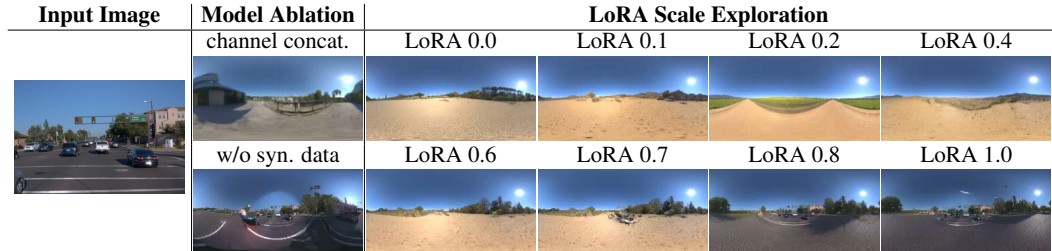

Figure 13: Model design ablation and LoRA scale exploration. The "Model Ablation" column shows the results of our two model design variants: 1) channel concatenation and 2) training without synthetic rendering data. The "LoRA Scale Exploration" columns show the visual results of our model with different LoRA scales.

| FOV | 45° | 50° | 60° | 70° | 75° |
|---|---|---|---|---|---|
| GT | | | | | |

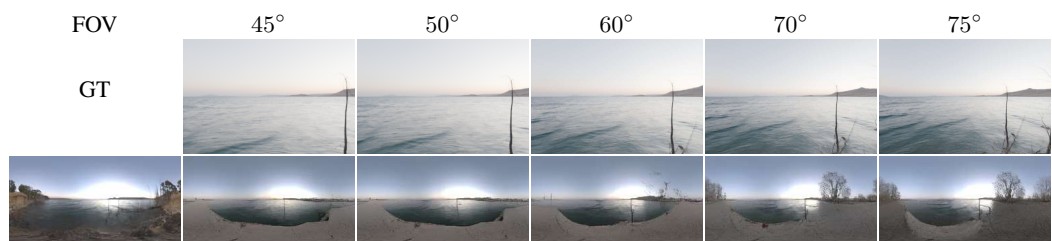

Figure 14: Lighting estimation from input images with varying camera FOV.

| Elevation | 30° ↑ | 15° ↑ | 0° | 15° ↓ | 30° ↓ |
|---|---|---|---|---|---|
| GT | | | | | |

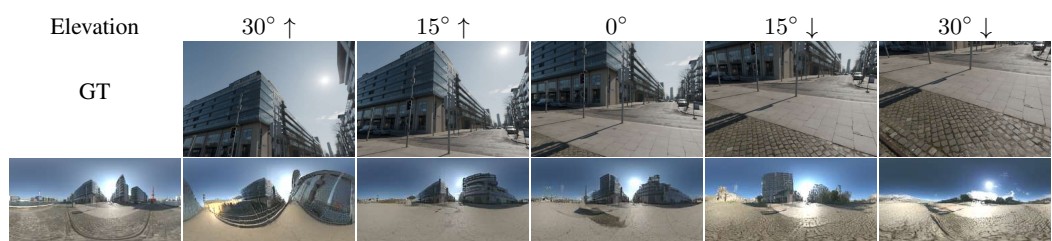

Figure 15: Lighting estimation from input images with varying camera elevation.

| Input Image | LoRA Scale Exploration | | |
|---|---|---|---|
| | GT | LoRA 0.00 | LoRA 0.25 |
| | LoRA 0.50 | LoRA 0.75 | LoRA 1.00 |

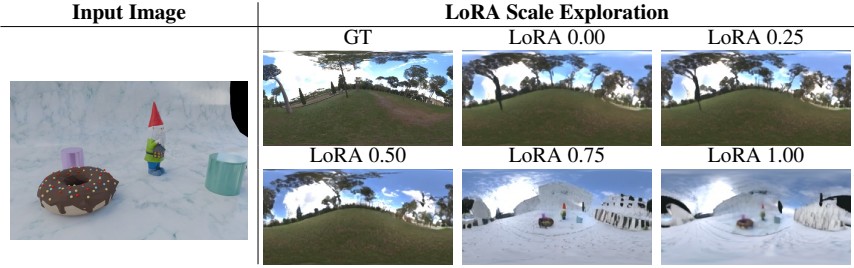

Figure 16: LoRA scale exploration on synthetic foreground scenes.

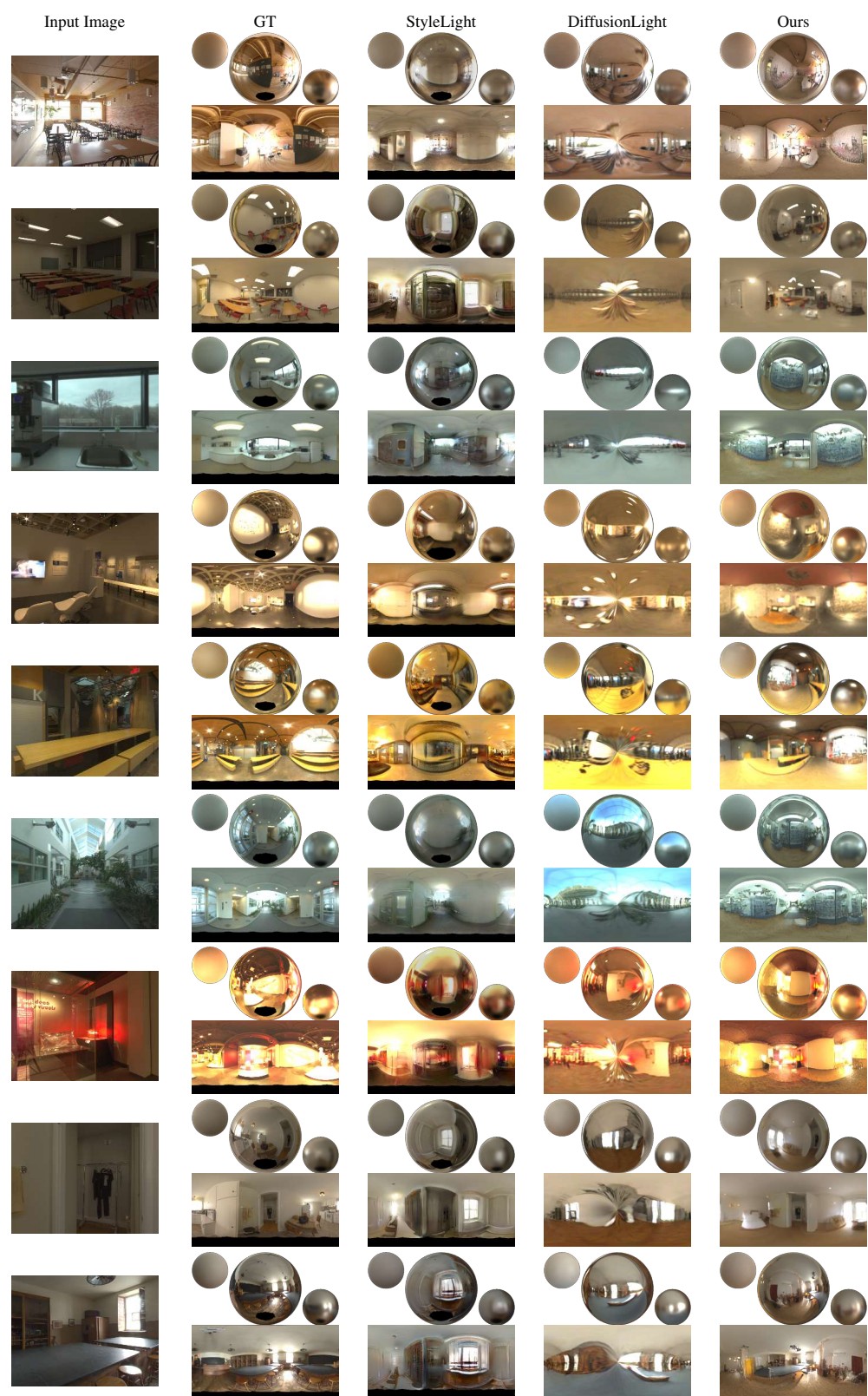

Figure 17: Additional qualitative results on Laval Indoor dataset.

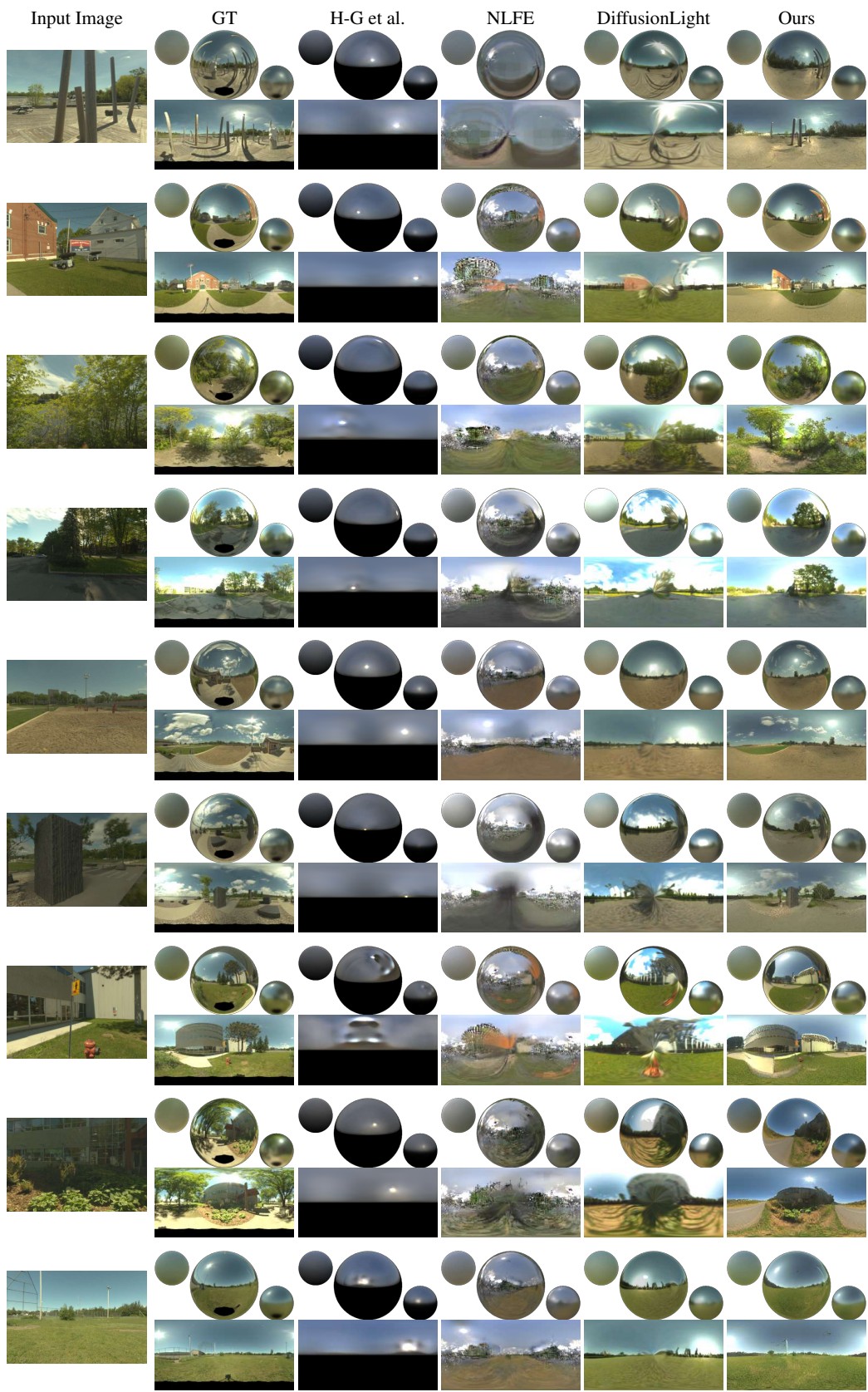

Figure 18: Additional qualitative results on Laval Outdoor dataset.

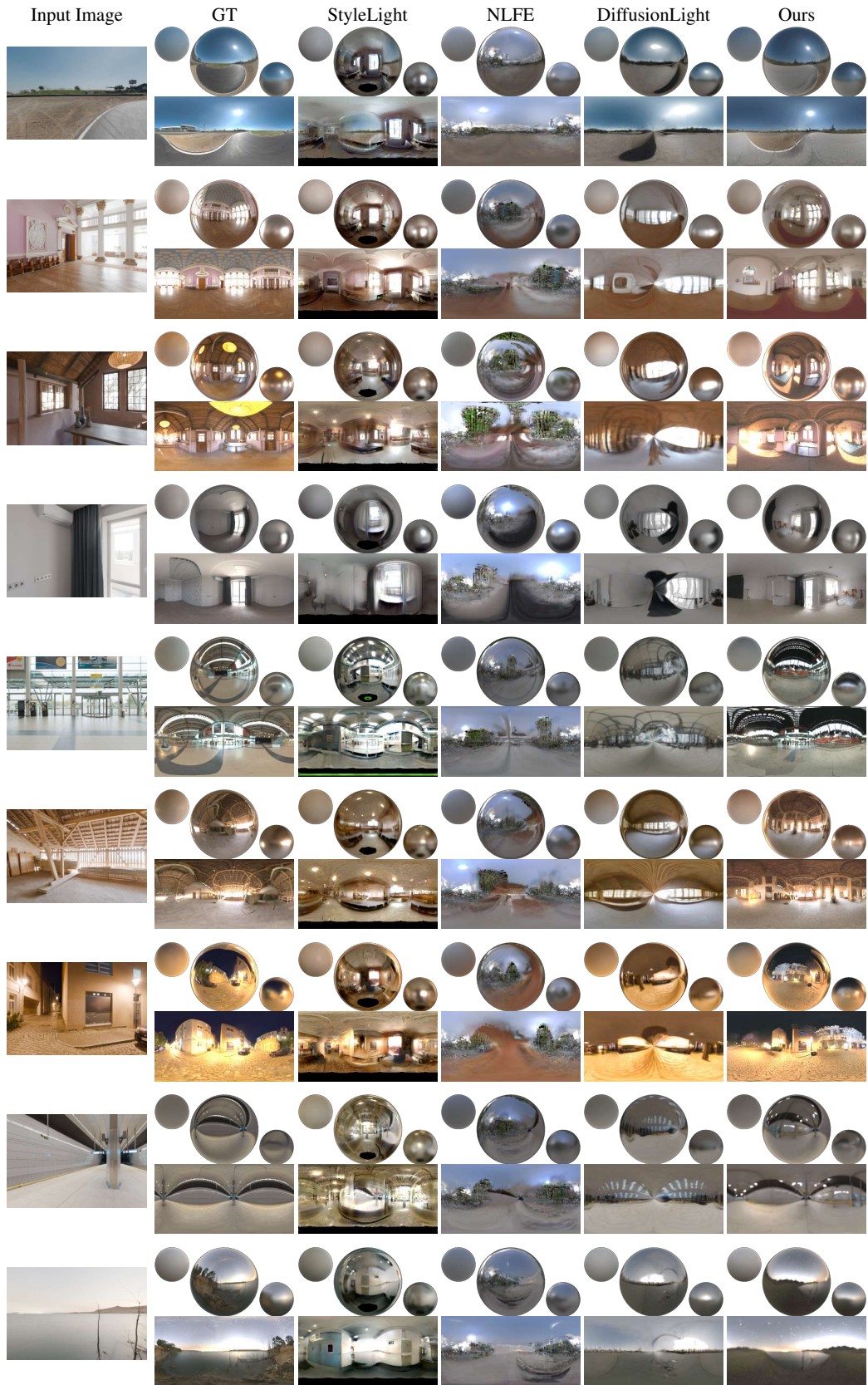

Figure 19: Additional qualitative results on Poly Haven dataset.

| Reference | StyleLight | DiffusionLight | DiPIR | Ours |
|-----------|-----------|----------------|-------|------|

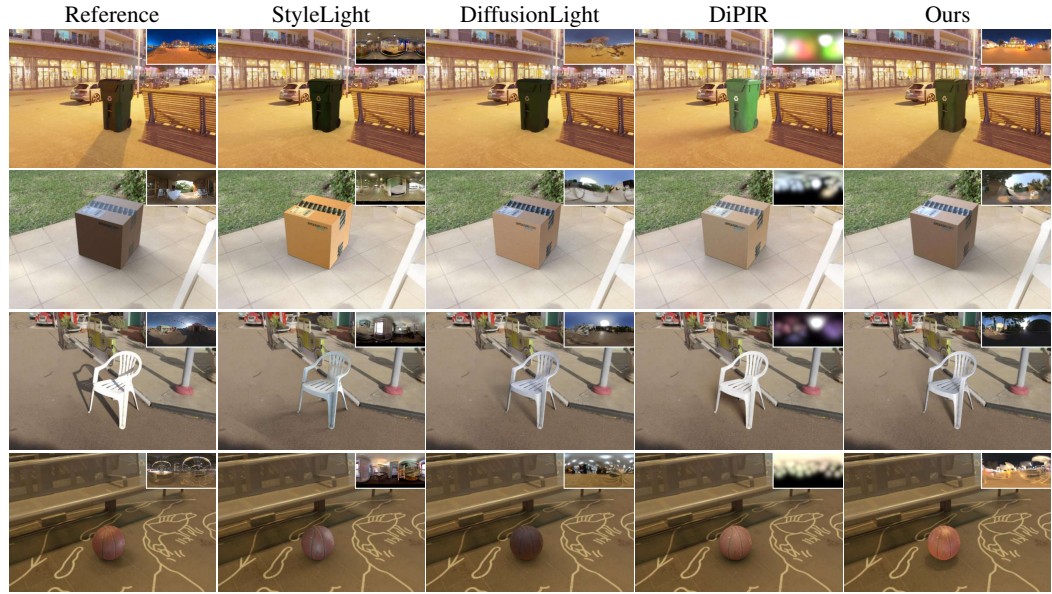

Figure 20: Additional virtual object insertion on Poly Haven perspective crops.

| H-G et al. [23] | NLFE | DiffusionLight | DiPIR | Ours |
|-----------------|------|----------------|-------|------|

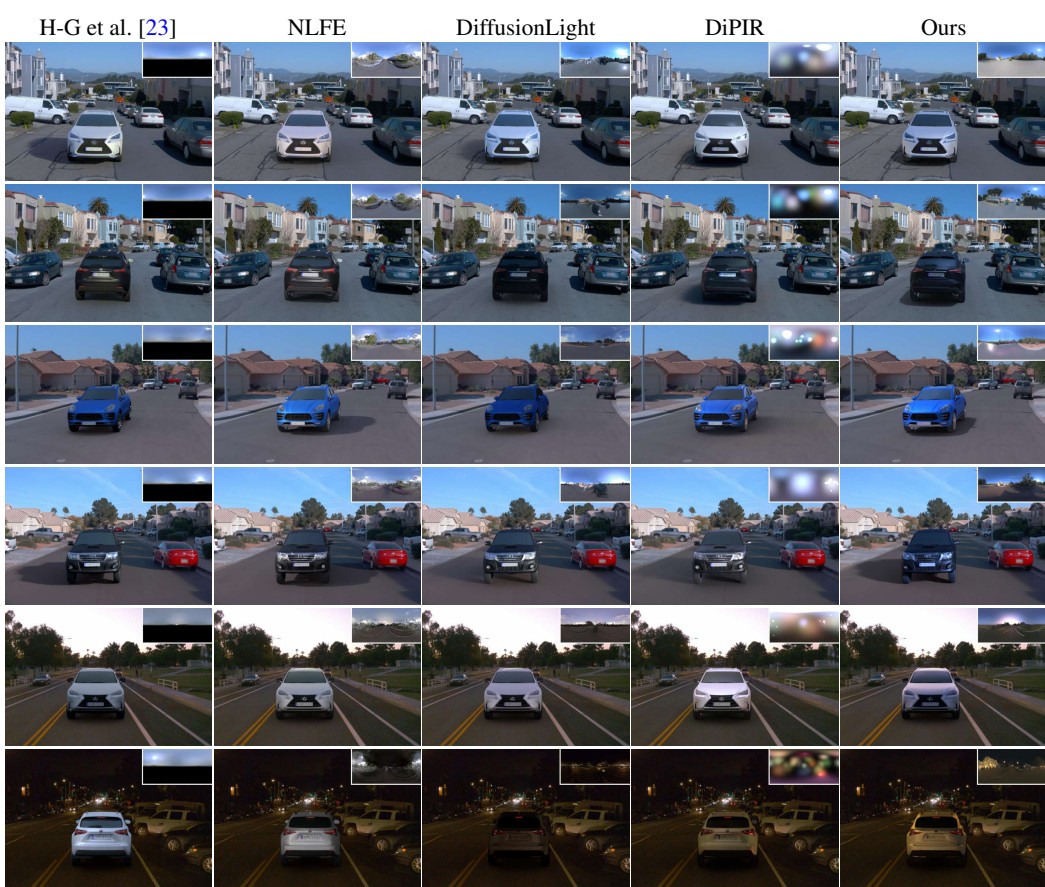

Figure 21: Additional virtual object insertion on Waymo driving scenes.

