# OpenReview forum: "LuxDiT: Lighting Estimation with Video Diffusion Transformer"
_NeurIPS.cc/2025/Conference — NeurIPS 2025 poster_

### Official Review · Reviewer_YCWS · 2025-06-01

**Clarity:** 3
**Significance:** 4
**Originality:** 4
**Rating:** 5
**Confidence:** 5

**Summary:**

This paper proposes LuxDiT, a method for estimating HDR environment maps from images or videos. It fine-tunes CogVideoX by replacing its text tokens with visual condition tokens derived from the environment map. To address the VAE's LDR limitation, the method uses dual tone-mapped representations (LDR and LogHDR) instead of raw HDR. A small MLP is then used to reconstruct the HDR map from these. Finally, LoRA fine-tuning with real-world LDR panoramas enhances semantic alignment. The method achieves state-of-the-art results in both qualitative and quantitative evaluations.

**Questions:**

1. **Stage II Input Representation:**
 In Equation 5 (Page 6, Line 214), what does $z_t$ represent during Stage II (semantic adaptation)? Since $z^{log}$ is unavailable for LDR-only inputs, how do you maintain the expected latent shape for the diffusion model? Equation 4 suggests the model expects a concatenated pair [$z^{ldr}$,$z^{log}$], which is shape [l,h,w,2c] but LDR-only inputs would be shape [l,h,w,c]

2. **VAE Usage with Single Images:**
 How do you apply the CogVideoX VAE, which compresses 4-frame video segments for single-frame inputs (ie, estimate lighting for image input)? Was the VAE modified, or is there a workaround to ensure compatibility?

3. **Directional Embedding Effectiveness:**
Directional embedding (Page 4, Line 156) is introduced to improve angular continuity, but no supporting ablation is provided. However, no evidence or ablation study is provided. Could you show results comparing models with and without this component?

4. **Inference Procedure:**
At inference time, when the $z^{ldr}$ and $z^{log}$ are initialized from Gaussian noise, is directional embedding still applied?

5. **Clarification of Figure 2:**
Figure 2 implies the latent shape becomes [l, h, w, 3C] through channel concatenation, but Lines 159–160 describe a channel-wise addition of E_dir to the token, suggesting the shape should be [l,h,w,2C]. Could you clarify this discrepancy?

**Ethical Concerns:**

["NO or VERY MINOR ethics concerns only"]

**Final Justification:**

I find the technical contribution of this paper—conditioning a video diffusion model to generate HDR environment maps—both novel and effective, with clear improvements over previous works.

The authors have satisfactorily addressed my concerns, which primarily focused on implementation details to ensure reproducibility. Based on the strength of the results and the clarified responses, I recommend acceptance.

**Limitations:**

Yes

**Paper Formatting Concerns:**

No concerns. The paper follows the NeurIPS template, stays within the 9-page content limit, and appears properly formatted.

**Quality:**

3

**Strengths And Weaknesses:**

## Strengths
- **Quality:**  Medium-quality work with solid experiments compared against state-of-the-art methods. The method of LuxDiT is technically sound. However, it lacks some key ablations (e.g., the effect of directional embedding).

 - **Significance:** The method shows significant improvements in lighting estimation, supported by strong quantitative and qualitative results.

 - **Originality:** The paper introduces 2 novel ideas
> 1. Use of a video diffusion transformer to estimate lighting from images or video.
> 2. Directional embedding to enhance angular continuity in the predicted environment map.

## Weakness
- **Clarity:** While the core idea is clear, the paper lacks sufficient implementation details. For example, the architecture of the MLP and the process of fine-tuning CogVideoX—especially how the model handles the removal of text tokens—are not adequately explained in the main paper. Some of this is covered in the appendix, but a brief summary or a clear reference in the main text would help. Additionally, Figure 2 seems inconsistent with the description in Lines 159–160: it suggests direct channel concatenation of $E_{ldr}$, $E_{log}$ and $E_{dir}$, whereas the text explain that $E_{ldr}$ and $E_{log}$ are concatenated and $E_{dir}$ is added via channel-wise addition.

---

> ### Author Rebuttal · Authors · 2025-07-31
>
> We thank the reviewer for their valuable feedback, and appreciate that they recognize our contributions. We respond to the comments and questions below.
>
> **W: Clarity of Implementation Details.** We appreciate the reviewer's feedback regarding implementation details. Supplements B.1 and B.3 provide detailed designs of our model, including the HDR MLP and modifications to CogVideoX. We will incorporate additional implementation details in the revised main paper.
>
> **W&Q5: Clarification of Figure 2.** We apologize for the inconsistent description in the draft regarding Figure 2. The concatenation shown in Figure 2 and the addition described on Lines 159-160 for directional latent conditioning are essentially equivalent. Specifically, in CogVideoX, before being processed by transformer blocks, the concatenated latents $[\mathbf{z}^{ldr}, \mathbf{z}^{ldr}, \mathbf{z}^{dir}]$ are processed by a projection layer: $\mathbf{h} = W \cdot [\mathbf{z}^{ldr}, \mathbf{z}^{ldr}, \mathbf{z}^{dir}]$. This equation can be re-formulated as $\mathbf{h} = W_{0:2C} \cdot [\mathbf{z}^{ldr}, \mathbf{z}^{ldr}] + W_{2C:3C} \cdot [\mathbf{z}^{dir}]$, which aligns with the description on Lines 159-160. To eliminate this confusion, we will correct this description in the revised paper.
>
> **Q1: Stage II input representation.** In Stage II, the noisy latent $\mathbf{z}_t$ is identical to that in Stage I. Initially, $\mathbf{z}_T$ is set to Gaussian noise with dimensions $[l, h, w, 2c]$. The DiT model continues to denoise this concatenated noisy latent $\mathbf{z}_t$ throughout Stage II training. The main distinction is that the denoised log part $\mathbf{z}^{log}_T$ is excluded from the diffusion loss computation if the input videos lack log maps.
>
> **Q2: VAE usage with single images.** When the single image is used as the input, CogVideoX’s 3D causal VAE will encode the single-frame image as special independent latent tokens (8x8x1 compression for the first frame). The original CogVideoX model is also jointly trained with single-frame image generation to support this.
>
> **Q3: Directional embedding effectiveness.** To enhance our model's ability to reason about the 3D orientation of environment maps, we utilize an additional directional embedding. This embedding allows us to augment our model training by randomly rolling or warping the input directional map, enabling the model to generate a corresponding environment map. Without this directional embedding, the model would only be able to predict environment maps with a fixed orientation, which could negatively impact its angular continuity. We additionally fine-tuned the model without using directional embedding and evaluated it on a subset of Polyhaven envmaps; the angular errors with three-sphere evaluation are reported in the table below, which justifies the effectiveness of directional embedding.
>
> |             | Diffuse | Matte | Mirror |
> | ----------- | ------- | ----- | ------ |
> | W/   Dir. | 1.273   | 1.979 | 2.967  |
> | W/o Dir.    | 1.481   | 2.420 | 3.501  |
>
> **Q4: Inference procedure.** At inference time, when $\mathbf{z}^{ldr}_T$ and $\mathbf{z}^{log}_T$ are initialized from Gaussian noise, we still concatenate the directional embedding along with denoising latents. However, the noise is not added to the directional embedding.

---

> > ### Comment · Reviewer_YCWS · 2025-08-02
> > **Implementation Details Clarified, I'm Satisfied with the Response**
> >
> > I had no concerns about the technical soundness of the paper; my questions focused on implementation details to ensure reproducibility. The authors have addressed them clearly, and I’m satisfied with their response.
> >
> >
> > Current Rating: 5 – Accept

---

### Official Review · Reviewer_DYHn · 2025-06-23

**Clarity:** 3
**Significance:** 4
**Originality:** 4
**Rating:** 5
**Confidence:** 4

**Summary:**

This paper presents LuxDiT, a DiT for HDR environmental lighting estimation from an LDR image or video. The model predicts LDR and Log-encoded environmental maps, and uses a separate MLP to fuse them back into the full HDR environment map. A small-scale LoRA finetuning stage with real-world HDR panoramas is added to improve real-world semantic consistency. The approach achieves state-of-the-art results on multiple benchmarks.

**Questions:**

1. Can you provide a direct comparison between your learned MLP HDR fusion and a simple rule-based baseline? As well as a CNN variant.
2. Clarify the concatenation method of z_log and z_ldr
3. Provide a quantitative evaluation of video temporal consistency
4. Redo the lora scale ablation on an object-centric dataset

**Ethical Concerns:**

["NO or VERY MINOR ethics concerns only"]

**Final Justification:**

Thank the authors for their rebuttal. The author's rebuttal clarified and resolved my concerns well, and I decided to keep my current positive rating.

**Limitations:**

yes

**Paper Formatting Concerns:**

No comments.

**Quality:**

3

**Strengths And Weaknesses:**

**Strengths**
1. The paper presents a clean and effective latent diffusion-based framework (DiT) for HDR panorama prediction, which is an important task, especially under today's trend that redoes everything with video generation models.
2. LoRA finetuning on real-world panorama crops improves semantic consistency; tone-mapping and exposure augmentations enhance robustness to in-the-wild image distributions, showing strong domain bridging ability.
3. The method achieves state-of-the-art performance on multiple public benchmarks.
4. The supplementary video shows qualitatively good temporal consistency on video inputs.

**Weaknesses**
1. **HDR Fusion**
    - The HDR fusion MLP is purely per-pixel and does not leverage spatial consistency, which may lead to local inconsistency.
    - No ablation is provided against rule-based HDR fusion methods like those used in DiffusionLight.
2. **Latent Concatenation Description Inconsistency**
    - Figure 2 may be misleading: Eldr, Elog, and Edir are encoded separately by the VAE; z_ldr and z_log are either channel- or token-wise concatenated, while z_dir is added.
    - Sec. 4.1 claims channel-wise concat, but Sec. 5.5 shows token-wise concat performs better. It's unclear which is actually used.
3. **Lack of Quantitative Temporal Analysis**
    - While qualitative results on video are shown, no quantitative metrics (e.g., angular jitter, HDR SSIM) are reported.
4. **LoRA Scale Ablation Unclear**
    - The LoRA scale ablation is inconclusive, as the finetuning data (PolyHaven) matches the test domain. It would be more informative to test degradation on other settings (e.g., synthetic or object-centric inputs).

---

> ### Author Rebuttal · Authors · 2025-07-31
>
> We thank the reviewer for their valuable feedback, and appreciate that they recognize our contributions. We respond to the comments and questions below.
>
> **W1&Q1: HDR fusion.** We found that per-pixel HDR fusion is sufficient for generating quality fused HDR maps. Our method exhibits fewer inconsistency issues compared to prior work, DiffusionLight. DiffusionLight estimates environment maps with varied exposure values through separate diffusion passes. In contrast, our method utilizes a single pass to obtain environment maps with two distinct tone-mappings. These two tone-mapped environment maps are decoded from different latent channels of the same denoised DiT tokens, which also enhances their spatial alignment.
>
> To justify our design choice, we compared various HDR fusion approaches, including MLP, CNN, and a rule-based method, to justify the use of a simple MLP. The rule-based method applies the inverted Reinhard map for lights with intensity below 8, a linear interpolation between Reinhard and log maps for intensities between 8 and 16, and exclusively the log map for intensities greater than 16. The table below shows the RMSE
>
> |      | MLP   | CNN   | Rule|
> | ---- | ----- | ----- | ---------- |
> | RMSE | 11.55 | 11.74 | 11.71      |
>
>
> All three approaches achieve a similar level of accuracy, with the MLP approach performing slightly better than the other two approaches. Compared to the rule-based approach, we believe the neural approach can better handle numerical inconsistency after image uint8 quantization, and the potential data range overflow (e.g., lights beyond the pre-defined maximum intensity 10000).
>
> **W2&Q2: Clarification of the concatenation method.** Our final model employs a dual concatenation approach. Specifically, the latents for $E_{ldr}$ , $E_{log}$ , and $E_{dir}$ are concatenated channel-wise. On the other hand, the latent for the input image condition $\mathbf{I}$ is token-wise concatenated with envmap tokens, as illustrated in Fig. 2. Regarding the ablation results in Table 7, Sec. 5.5, the "channel concat." variant concatenates all noisy and conditional latents along the channel dimension. For this variant, the envmap $\mathbf{E}$ is resized to match the input image $\mathbf{I}$, and no token-wise concatenation is used. To prevent confusion and misunderstanding, we will further clarify this design and enhance the consistency of our descriptions in the revised paper.
>
> **W3&Q3: Quantitative temporal analysis.** We provided the metrics for measuring temporal consistency in Table 3. We computed the standard deviation (std) of per-frame error metrics for each video clip (L268-L270). For example, “PAE Std” in Table 3 measures the jittering scale of the peak light position in a video sequence.
>
> **W4&Q4: LoRA Scale Ablation.** As suggested by the reviewer, we made an ablation on the LoRA Scale with input synthetic objects. The angular errors with three-sphere evaluation are shown in the table below
>
> | LoRA Scale   | Diffuse | Matte | Mirror |
> | ----- | ------- | ----- | ------ |
> | 0.00 | 1.594   | 2.068 | 3.405  |
> | 0.25 | 1.737   | 2.311 | 3.690  |
> | 0.50 | 2.170   | 2.914 | 4.342  |
> | 0.75 | 3.832   | 5.322 | 6.783  |
> | 1.00 | 3.937   | 5.891 | 7.400  |
>
> Contrary to the ablation made on scene images (Polyhaven crops), as the LoRA scale becomes larger, the corresponding lighting estimation accuracy becomes lower. We will also include visual results in the revised paper.

---

> > ### Comment · Reviewer_DYHn · 2025-08-04
> >
> > Thank the authors for their rebuttal. The author's rebuttal clarified and resolved my concerns well, and I decided to keep my current positive rating.

---

### Official Review · Reviewer_GYdA · 2025-06-29

**Clarity:** 3
**Significance:** 2
**Originality:** 2
**Rating:** 3
**Confidence:** 4

**Summary:**

This paper presents LuxDiT, a transformer-based diffusion framework for estimating HDR environment maps from a single image or video. The model is built on top of a video diffusion transformer and operates on a dual-tonemapped representation of HDR lighting (Reinhard and log-tonemapped variants), which is jointly denoised in latent space. A novel conditioning mechanism, directional embedding, and two-stage training (synthetic pretraining followed by LoRA-based adaptation) are introduced. Evaluation is conducted on three benchmarks, showing improved performance over prior methods like StyleLight and DiffusionLight.

**Questions:**

-There’s no discussion of why these baselines were excluded, and as a result, the evaluation appears narrow and potentially cherry-picked.
-How camera parameters are selected for the LDR crops.
-How robustness to real-world scenes is assessed.
-Whether the model generalizes to truly novel lighting conditions (e.g., high-contrast outdoor settings).

**Ethical Concerns:**

["NO or VERY MINOR ethics concerns only"]

**Final Justification:**

Although the concerns of experiments and some clarification have been well addressed, I still feel the novelty is an issue. But I won't argue for rejecting this paper.

**Limitations:**

The major limitation is that improvements are not significant.

**Paper Formatting Concerns:**

No.

**Quality:**

2

**Strengths And Weaknesses:**

Strengths:
-HDR environment map estimation from a single image or video is a challenging and important task for applications in graphics, relighting, and AR/VR.
-The use of dual tonemapping (Reinhard + log) for representing HDR panoramas and the joint transformer denoising approach is technically interesting.

Weaknesses:
-While the architecture is inspired by recent diffusion transformers and introduces some adaptations (e.g., token-based conditioning, direction maps), these are incremental and follow standard practices (e.g., AdaLN, LoRA fine-tuning).
The novelty of combining two tonemapped HDR representations is not sufficiently justified. Prior works (e.g., in inverse rendering or HDR synthesis) have used log-space representations extensively. The added complexity of dual-tonemapping is not ablated clearly.
-The reported improvements are small and inconsistent. For instance, on the Laval Indoor dataset, the model slightly improves on RMSE but does not outperform prior work on angular error across the board. Some results are within statistical noise and lack significance testing or confidence intervals.
-Missing Evaluation on Key Baselines: The baselines compared against (StyleLight, DiffusionLight) are relevant, but the paper does not consider broader categories of lighting estimation models—e.g., inverse rendering models or methods like NeRFactor, InverseRenderNet, or relighting-specific diffusion models.

---

> ### Author Rebuttal · Authors · 2025-07-31
>
> We thank the reviewer for their valuable feedback, and appreciate that they recognize our contributions. We respond to the comments and questions below.
>
> **W1: Technical contribution.** We thank the reviewer for prompting a deeper discussion of our contributions.
>
> LuxDiT tackles two persistent challenges in HDR lighting estimation: (1) predicting accurate intensity and direction from spatial inputs, which requires inferring global, non-spatial outputs (360° panoramas) from local, spatial visual cues; and (2) recovering angular high-frequency details consistent with the scene content.
>
> Our solution combines: (1) a scalable generative architecture that leverages a powerful pre-trained video diffusion model and a carefully curated large-scale synthetic dataset; (2) dual-tonemapped latent representations for HDR structure; and (3) LoRA fine-tuning on real HDR panoramas to improve semantic alignment. While each component builds on existing techniques, their integration is purposefully designed for the unique demands of lighting estimation.
>
> To our knowledge, LuxDiT is the first end-to-end training method to formulate lighting prediction as a conditional generative diffusion task. We conduct extensive evaluations and ablations. LuxDiT reduces lighting direction error by 45% on Laval Outdoor (Table 2) and improves temporal consistency on video input (Table 3), enabling reliable use in downstream tasks such as virtual object insertion. We will release our models, dataset, and code to support reproducibility and future work.
>
> **W1: Dual-Tonemapping Light Representation.** The use of multiple tonemapped light maps for HDR representation is common in rendering tasks. Similar approaches have been applied in prior methods for both lighting estimation (e.g., DiffusionLight) and relighting tasks (e.g., Neural Gaffer).
>
> An alternative choice for our HDR light representation is to follow DiffusionLight’s multi-pass, multi-EV tonemapping. However, this design choice also falls short of our requirement for higher dynamic range environment maps with spatially coherent content.
> DiffusionLight's reliance on separate diffusion passes for estimating environment maps with varied exposure values introduces spatial inconsistencies. Its multi-EV HDR fusion results in a limited dynamic range, with a maximum intensity of 32.
> In contrast, our proposed method, LuxDiT, addresses these limitations by jointly denoising dual-tonemapped environment maps within the same latent tokens. This approach significantly enhances spatial consistency during HDR fusion. Furthermore, LuxDiT utilizes Reinhard and logarithm tonemapping functions, which provide sufficient sampling to cover a broad range of lighting scenarios, from rich LDR content to high-contrast spotlights (as shown in Supplementary Fig. S1). This enables HDR reconstruction with a peak intensity of up to 10,000.
>
>
> **W2: Significance of quantitative improvements.** We evaluated our method on various benchmark datasets with different evaluation metrics. Specifically, on Laval Outdoor scenes, our model substantially enhances peak light direction estimation quality, reducing the error by 45% (Table 2). Furthermore, it improves video lighting consistency by decreasing temporal jittering by up to 81% (Table 3, "std" terms).
>
> The reviewer's concern regarding metric inconsistency primarily stems from the Laval Indoor experiment. As stated in the paper (L242-L245), our model was not trained on the Laval Indoor dataset. This dataset exhibits a notable shift in color and intensity distribution compared to our training set. In contrast, the baseline method, StyleLight, is exclusively trained on Laval Indoor. While DiffusionLight does not directly use Laval Indoor, the Text2Light model, which generates its training data, is trained with Laval Indoor data. Despite these factors, our method demonstrates strong robustness by still achieving competitive metrics on the Laval Indoor dataset.
>
> Regarding statistical noise and confidence interval, as also mentioned by R1, we evaluated LuxDiT three times using different random seeds and reported the average metrics. For brevity, we omitted the standard deviations in the main paper.
> Below, we provide the standard deviations for the metrics reported in Table 1 of the main paper.
> As shown in the table, the variance of metrics is small enough, not affecting the comparison results shown in the paper.
>
> |               | si-RMSE      |              |              | Angular Error |              |              | norm-RMSE    |              |              |
> | ------------- | ------------ | ------------ | ------------ | ------------- | ------------ | ------------ | ------------ | ------------ | ------------ |
> | Dataset       | Diffuse      | Matte        | Mirror       | Diffuse       | Matte        | Mirror       | Diffuse      | Matte        | Mirror       |
> | Laval Indoor  | 0.112±0.001 | 0.297±0.007 | 0.586±0.011 | 2.555±0.084  | 3.526±0.092 | 5.641±0.114 | 0.196±0.011 | 0.341±0.010 | 0.457±0.019 |
> | Laval Outdoor | 0.068±0.002 | 0.190±0.005 | 0.396±0.014 | 2.018±0.081  | 2.939±0.018 | 5.286±0.021 | 0.137±0.006 | 0.271±0.009 | 0.454±0.020 |
> | Poly Haven    | 0.077±0.003 | 0.196±0.008 | 0.442±0.009 | 1.235±0.061  | 1.977±0.058 | 2.783±0.058 | 0.111±0.005 | 0.199±0.004 | 0.323±0.003 |
>
>
> **W3&Q1: Evaluations on other key baselines.** We thank the reviewer for highlighting broader categories of lighting-related methods. Our work focuses specifically on single-view, geometry-free estimation of scene-scale 360° HDR environment maps.
>
> Methods like NeRFactor operate under different assumptions. They typically require multi-view posed images and geometry to optimize lighting, shape, and material, and are designed for object-centric, per-scene optimization. Our method, conversely, focuses on generalizable, image-conditioned lighting prediction, making direct comparisons on standard lighting estimation benchmarks (e.g., Laval, PolyHaven) infeasible due to the distinct input conditions and scale of light estimation (object vs. scene).
>
> We thank the reviewer for mentioning inverse rendering models for the lighting estimation. We did not deliberately exclude the inverse rendering models in our evaluation. The input conditions for inverse rendering models and image-based lighting estimation are very different. Inverse rendering requires additional multi-view posed images and 3D geometry for estimating light probes, and such light estimation is usually on object scale instead of scene scale. It is very difficult for us to report inverse rendering models’ metrics in Table 1.
>
> That said, we include preliminary comparisons with inverse rendering methods (e.g., NVDIFFREC and NVDIFFRECMC) in Supplementary Sec. C.2 (Fig. S5). We also evaluated our method on the Objects-with-Lighting dataset [1] against NeuS+Mitsuba, as reported in the table below.
>
> |              | si-RMSE |       |        | Angular | Error     |        | norm |   RMSE    |        |
> | ------------ | ------- | ----- | ------ | ------------- | ----- | ------ | --------- | ----- | ------ |
> | Method       | Diffuse | Matte | Mirror | Diffuse       | Matte | Mirror | Diffuse   | Matte | Mirror |
> | NeuS+Mitsuba | 0.082   | 0.232 | 0.424  | 3.145         | 3.383 | 3.526  | 0.180     | 0.545 | 0.717  |
> | Ours         | 0.086   | 0.253 | 0.482  | 1.262         | 1.594 | 2.000  | 0.153     | 0.339 | 0.479  |
>
> Our method is not expected to completely surpass inverse rendering approaches. Instead, we believe that integrating our generative model with inverse rendering methods could further improve lighting estimation.
>
> The reviewer mentioned InverseRenderNet, an inverse rendering model that estimates lighting using low-frequency spherical harmonics (SH) from a single image. Our paper compares against models that also predict low-frequency lighting from RGB input (e.g., H-G et al. in Tables 1 & 2), and LuxDiT surpasses their performance. EMLight [2] is another method that relies on low-frequency light as guidance; the StyleLight paper included EMLight in its comparison, and EMLight did not outperform StyleLight. A comprehensive evaluation of earlier methods like InverseRenderNet is beyond the scope and time constraints of this rebuttal, but we will incorporate a discussion of its relevance to our work in the revised version.
>
> **Q2: Camera parameters for LDR crops.** We uniformly sample the FOV of the perspective camera within the range between 45 and 80 degrees. The camera elevation angle is sampled between -10 and +10 degrees.
>
> **Q3&Q4: Robustness and generalization to real-world scenes.** In the paper, we used the virtual object insertion rendering to show our model’s robustness and generalization to real-world scenes. We show the metrics in Table 8, visual rendering results in Fig. 1, Fig. 4, and Supplementary Fig. S11, S12. Note that all the Waymo driving scenes can be treated as in-the-wild real-world outdoor captures (not cropped from any panorama). Our model effectively estimates high-contrast sunlight, allowing for the accurate casting of sharp shadows from inserted objects.
>
>
> > [1] Ummenhofer, Benjamin, et al. "Objects with lighting: A real-world dataset for evaluating reconstruction and rendering for object relighting." 3DV, 2024.
>
> > [2] Zhan, Fangneng, et al. "Emlight: Lighting estimation via spherical distribution approximation." AAAI, 2021.

---

> ### Comment · Reviewer_YCWS · 2025-08-02
> **Pointing out additional novel contribution**
>
> I agree with Reviewer GYdA30 that dual-tonemapping is not a novel contribution of this work.
>
> However, I believe the proposed method for conditioning a diffusion model to generate artifact-free environment maps—particularly avoiding the common center artifacts seen in DiffusionLight—is a valuable contribution. This improvement is evident in Figure 3 of the paper and in comparison to qualitative results from the DiffusionLight paper.

---

> ### Comment · Reviewer_GYdA · 2025-08-05
>
> Thanks for the rebuttal. My concerns have been addressed. But the novelty is still not convincing to me. I will raise the points up to 3.

---

### Official Review · Reviewer_b4Ep · 2025-07-05

**Clarity:** 3
**Significance:** 2
**Originality:** 2
**Rating:** 4
**Confidence:** 4

**Summary:**

The paper proposes LuxDiT, a novel method for estimating high-dynamic-range (HDR) lighting from single images or videos using a video diffusion transformer (DiT). The model is trained on a large-scale synthetic dataset and then adapted to real-world scenes using LoRA-based fine-tuning on curated HDR panoramas. The approach demonstrates state-of-the-art results in both image- and video-based lighting estimation tasks and downstream applications such as virtual object insertion.

**Questions:**

1. Can you provide more details on new provided dataset?
2. Can you demonstrate your approach using TCC or Cube++ dataset?
3. Can you show variances of errors in your experimental table?
4. Does your improvement positively affects on final renedring quality?

**Ethical Concerns:**

["NO or VERY MINOR ethics concerns only"]

**Final Justification:**

The main point appers to be very convincing about paper valuability is a demo of the results. It is really nice to see significant improvement in this very complicated problem. I still feel that this paper have issues with physical side of the problem, yet neural side looks pretty good

**Limitations:**

1. Very high computational time
2. Not clearly described issues which have been significantly improved with new approach

**Paper Formatting Concerns:**

I haven't notices any issues with paper formatting

**Quality:**

3

**Strengths And Weaknesses:**

Strengths:
Paper written with clear motivation has good structure and reasoning.

1, Authors demonstrate the power of modern diffusion model and showed possibility to proivide good prediction of HDR maps.
2. Exhaustive work with existing dataset helps to verify that authors achieves a quality improvement. Existing SotA methods are also mentioned.
3. Authors provides a good dataset for highlighted problem, which has a good potential to attract more attention to the problem

Weaknesses:
1. My main concern is that this paper has only incremental improvement. It is hard to understand from the paper answer on the question: does the authors achieved solution of some existing issue? May be in scope of the photorealistic rendering some user study would help to highlight significance of the result.
2. Experimental study a bit indirectly indicate improvements, moreover authors didn't provide variance of errors, so it is impossible to evalute the significance of improvement
3. Not all the propper dataset have been used for the problem. I recommend to use TCC and Cube++, since they also have mirror ball in the scene

---

> ### Author Rebuttal · Authors · 2025-07-31
>
> We thank the reviewer for their valuable feedback, and appreciate that they recognize our contributions. We respond to the comments and questions below.
>
> **W1: Contribution.** We thank the reviewer for prompting a deeper discussion of our contributions.
>
> LuxDiT tackles two persistent challenges in HDR lighting estimation: (1) predicting accurate intensity and direction from spatial inputs, which requires inferring global, non-spatial outputs (360° panoramas) from local, spatial visual cues; and (2) recovering angular high-frequency details consistent with the scene content.
>
> Our solution combines: (1) a scalable generative architecture that leverages a powerful pre-trained video diffusion model and a carefully curated large-scale synthetic dataset; (2) dual-tonemapped latent representations for HDR structure; and (3) LoRA fine-tuning on real HDR panoramas to improve semantic alignment. While each component builds on existing techniques, their integration is purposefully designed for the unique demands of lighting estimation.
>
> To our knowledge, LuxDiT is the first end-to-end training method to formulate lighting prediction as a conditional generative diffusion task. We conduct extensive evaluations and ablations. LuxDiT reduces lighting direction error by 45% on Laval Outdoor (Table 2) and improves temporal consistency on video input (Table 3), enabling reliable use in downstream tasks such as virtual object insertion. We will release our models, dataset, and code to support reproducibility and future work.
>
> **W1&Q4: Rendering quality evaluation with estimated lighting.** We use the lighting estimated by LuxDiT  to render virtual objects for insertion into real scenes, as described in the main paper Sec. 5.4. Qualitative results are shown in Fig. 1, Fig. 4, and Supplementary Figs. S11 and S12. In response to reviewer W1, a user study was also conducted to evaluate perceptual quality (further details in Supp. Sec. B.4). As shown in Table 8, renderings generated with LuxDiT's lighting were favored over those produced by baseline methods.
>
> **W2&Q3: Variance of metrics.** We thank the reviewer for pointing out the lack of variance reporting. Due to the stochastic nature of diffusion models, we evaluated LuxDiT three times using different random seeds and reported the average metrics. For brevity, we omitted the standard deviations in the main paper. Below, we provide the standard deviations for the metrics reported in Table 1 of the main paper.
>
> |               | si-RMSE      |              |              | Angular | Error       |              | norm-RMSE  |           |              |
> | ------------- | ------------ | ------------ | ------------ | ------------- | ------------ | ------------ | ------------ | ------------ | ------------ |
> | Dataset       | Diffuse      | Matte        | Mirror       | Diffuse       | Matte        | Mirror       | Diffuse      | Matte        | Mirror       |
> | Laval Indoor  | 0.112±0.001 | 0.297±0.007 | 0.586±0.011 | 2.555±0.084  | 3.526±0.092 | 5.641±0.114 | 0.196±0.011 | 0.341±0.010 | 0.457±0.019 |
> | Laval Outdoor | 0.068±0.002 | 0.190±0.005 | 0.396±0.014 | 2.018±0.081  | 2.939±0.018 | 5.286±0.021 | 0.137±0.006 | 0.271±0.009 | 0.454±0.020 |
> | Poly Haven    | 0.077±0.003 | 0.196±0.008 | 0.442±0.009 | 1.235±0.061  | 1.977±0.058 | 2.783±0.058 | 0.111±0.005 | 0.199±0.004 | 0.323±0.003 |
>
> As shown in the table above, the variance of metrics is small enough, not affecting the comparison results shown in the paper.
>
> **W3&Q2: Additional evaluation on TCC or CUBE++.** We thank the reviewer for suggesting additional datasets for evaluation. We performed an additional experiment using 100 images from Cube++. We applied both DiffusionLight and our method to estimate lighting from each image, then rendered the left and right white faces of the SpyderCube under the estimated illumination, assuming purely diffuse reflectance. We compared the rendered face colors to the provided ground truth and reported RMSE and angular error. Our method outperforms DiffusionLight across all reported metrics.
>
> We note that TCC and Cube++ are primarily designed for color constancy rather than 360° HDR lighting estimation, and the reflective spheres in these datasets lack accurate HDR ground truth or consistent annotations. As such, this experiment is inherently approximate and should be considered a supplementary evaluation.
>
> |                | RMSE       |            | Angular    | Error      |
> | -------------- | ---------- | ---------- | ----------- | ----------- |
> |                | Left       | Right      | Left        | Right       |
> | DiffusionLight | 0.115      | 0.112      | 18.648      | 18.379      |
> | Ours           | **0.110** | **0.110** | **18.066** | **18.050** |
>
> **Q1: More details on our dataset.** We thank the reviewer for their interest in our provided dataset. Details of the dataset curation process, including selection criteria, rendering setup, and preprocessing steps, are provided in Supplementary Sec. B.2. This dataset will also be released.

---

> > ### Comment · Reviewer_b4Ep · 2025-08-07
> > **Doubts about result on Cube++ and ignoring mirror ball**
> >
> > After some time thinking and googling I became more doubtful about the result and metric in the paper.
> > In fact angular error > 15 degrees is drastically larger than even GrayWorld. Having this observation in mind, I have trhee point of discussion:
> > - How do you think, why do we have such large error?
> > - Does it mean that errors used in the paper is not informative enough and do not indicate quality of illumination estimation? since LuxDit reconstruct an image it might be overfitted to predict overage scene structure, which would be close to GT, but wont have connection with illumination parameters. Is there any way to verify this?
> > - In fact there is no lack in markup of mirror ball for Cube++, I've googled it: https://github.com/createcolor/IDE/tree/develop/markup

---

> ### Author Response · Authors · 2025-08-08
>
> We appreciate the reviewer’s pointer to Cube++ annotations and the additional resources. In our rebuttal-phase Cube++ experiment, the unexpectedly high angular errors for both LuxDiT and DiffusionLight were caused by a mismatch between the ground truth face colors we used and the JPEG images we tested on.
>
> Specifically, we mistakenly sampled ground truth values from `Cube++/auxiliary/extra/gt_json/*`, which are derived from RAW images, whereas our methods (and baseline DiffusionLight) operate on processed JPEG images with different color responses.
>
> For example, in Cube++ image 00_0001, the ground truth left-face color from the RAW-based JSON is (0.18, 0.46, 0.35), while the corresponding JPEG face color is approximately (0.32, 0.33, 0.35). LuxDiT’s rendering for that face is (0.30, 0.33, 0.37), yielding an angular error of 17.10° using RAW-based colors, but only 2.80° when compared correctly to JPEG-based colors.
>
> We have re-evaluated using the correct color sampled from JPEG image with face annotations from `Cube++/auxiliary/source/JPG.JSON/*`, without re-running the model. The corrected metrics are shown below. Both methods now achieve much lower angular errors, with LuxDiT clearly outperforming DiffusionLight, achieving <5° on both faces.
>
> |                | RMSE            |                 | Angular        | Error           |
> | -------------- | --------------- | --------------- | ---------------- | ---------------- |
> |                | Left            | Right           | Left             | Right            |
> | DiffusionLight | 0.0436           | 0.0352           | 7.221           | 5.741           |
> | Ours           | **0.0245** | **0.0247** | **3.985** | **4.003** |
>
>
> We thank the reviewer for bringing this to our attention. We will include the corrected evaluation in the revised supplementary material and discuss it in the main paper.

---

### Decision · Program_Chairs · 2025-09-17

**Decision:**

Accept (poster)

**Comment:**

This paper presents a framework for HDR panorama prediction based on a latent diffusion transformer. The approach utilizes LoRA finetuning on real-world panorama crops to ensure semantic consistency and applies tone-mapping and exposure augmentations to enhance robustness, resulting in strong domain bridging capabilities. The method achieves SOTA results across multiple benchmarks, with supplementary materials demonstrating impressive qualitative performance and temporal consistency.

While some concerns were raised regarding physical realism and limited novelty, these issues were adequately addressed in the rebuttal and during discussion. Reviewers generally agree that the technical contribution of conditioning a video diffusion model for HDR environment maps is both novel and effective.

Given the consistent support from reviewers, the strong empirical performance, and the demonstrated practical impact, the consensus is to accept the paper, and AC agrees with the majority.